# Integration of Ligand-Based and Structure-Based Methods for the Design of Small-Molecule TLR7 Antagonists

**DOI:** 10.3390/molecules27134026

**Published:** 2022-06-23

**Authors:** Sourav Pal, Uddipta Ghosh Dastidar, Trisha Ghosh, Dipyaman Ganguly, Arindam Talukdar

**Affiliations:** 1Department of Organic and Medicinal Chemistry, CSIR-Indian Institute of Chemical Biology, 4 Raja S. C. Mullick Road, Kolkata 700032, India; uddipta.ghoshdastidar@gmail.com (U.G.D.); trishaghosh146@gmail.com (T.G.); 2Academy of Scientific and Innovative Research, Ghaziabad 201002, India; 3IICB-Translational Research Unit of Excellence, Department of Cancer Biology and Inflammatory Disorders, CSIR-Indian Institute of Chemical Biology, CN6, Sector V, Salt Lake, Kolkata 700091, India; dipyaman@iicb.res.in

**Keywords:** drug design, QSAR, pharmacophore model, molecular dynamics, TLR7 antagonists

## Abstract

Toll-like receptor 7 (TLR7) is activated in response to the binding of single-stranded RNA. Its over-activation has been implicated in several autoimmune disorders, and thus, it is an established therapeutic target in such circumstances. TLR7 small-molecule antagonists are not yet available for therapeutic use. We conducted a ligand-based drug design of new TLR7 antagonists through a concerted effort encompassing 2D-QSAR, 3D-QSAR, and pharmacophore modelling of 54 reported TLR7 antagonists. The developed 2D-QSAR model depicted an excellent correlation coefficient (R^2^_training_: 0.86 and R^2^_test_: 0.78) between the experimental and estimated activities. The ligand-based drug design approach utilizing the 3D-QSAR model (R^2^_training_: 0.95 and R^2^_test_: 0.84) demonstrated a significant contribution of electrostatic potential and steric fields towards the TLR7 antagonism. This consolidated approach, along with a pharmacophore model with high correlation (R_training_: 0.94 and R_test_: 0.92), was used to design quinazoline-core-based hTLR7 antagonists. Subsequently, the newly designed molecules were subjected to molecular docking onto the previously proposed binding model and a molecular dynamics study for a better understanding of their binding pattern. The toxicity profiles and drug-likeness characteristics of the designed compounds were evaluated with in silico ADMET predictions. This ligand-based study contributes towards a better understanding of lead optimization and the future development of potent TLR7 antagonists.

## 1. Introduction

The innate immune system serves as the initial line of nonspecific evolutionary defensive approach against the entrance of pathogenic microorganisms. The innate immune system is a complex and ancient system of defense that includes pathogen recognition using germ-line-encoded pathogen receptors. Toll-like receptors (TLRs) are evolutionarily conserved pattern recognition receptors that play a pivotal role in sensing the invading pathogens by regulating inflammation [1,2,3,4]. In both healthy and disease conditions, these innate receptors play a significant role in identifying conserved pathogen-associated molecular patterns (PAMPs) associated with microbes, viz. bacteria, and viruses, as well as subsequently activating downstream pathways, resulting in innate immune responses [2,5,6]. The majority of the TLR family members are expressed on the surfaces of various immune cell subsets, except for TLR7, TLR8, and TLR9, which are found in the acidic environments (pH = 4.5 to 6.5) of endosomal compartments [7,8,9,10]. In the steady-state, TLR7 is exclusively expressed by plasmacytoid dendritic cells and B lymphocytes in humans [11].

TLRs are transmembrane proteins composed of 1049 amino acids and divided into three different structural domains: a ligand-binding ectodomain (ECD) with 27 leucine-rich repeats (LRRs) motifs, a cytoplasmic TIR (Toll/IL-1 receptor) domain responsible for downstream signalling, and a transmembrane domain that links the extracellular LRR ectodomain with the intracellular TIR domains. Unlike TLR8, which appears as a dimeric form [12], TLR7 exists as a monomer in the absence of ligands but is converted into a homodimeric arrangement in the presence of ligands. Another important conserved structural domain among TLR7, TLR8, and TLR9 is the Z-loop region (between the LRR14 and LRR15 interface of the ectodomain), which is important for homodimeric formation. TLR7, TLR8, and TLR9 are endosomal TLRs that specialize in recognizing nucleic acids of both pathogen origins acquired from phagocytosed microorganisms on their entry into acidic (pH < 6.5) endolysosomal compartments [1,5]. Both TLR7 and TLR8, which are structurally very similar, share the same immuno-cellular niche, and both recognize single-stranded RNA, while TLR9 recognizes unmethylated CpG DNA motifs. Upon activation, endosomal TLRs trigger a common cascade of downstream signalling pathways that results in the activation of NFκB transcription factors or mitogen-activated protein kinases (MAPKs), followed by the expression of different proinflammatory cytokines and IFN regulatory factors (IRFs) necessary for eventual recruitment functioning for acquired immunity [1,3,5,11,13,14].

Interestingly, different groups have also established that aberrant or over-endosomal TLR activation in response to self-nucleic acids, which can access the endosomal compartments of pDCs in different contexts (mostly in cases of the death of host cells and the subsequent extracellular release of nucleic acids in different forms), is a crucial pathogenic event in a wide range of autoimmune and metabolic disorders, including psoriasis [15,16], systemic lupus erythematosus (SLE) [17], scleroderma, rheumatoid arthritis (RA) [18], Sjogren’s syndrome [19], type 1 diabetes [20], type 2 diabetes [21], etc., all of which are characterized by extensive tissue damage. Revelations have been made about discoveries of pathological functions of endosomal TLRs in several chronic inflammatory and autoimmune disorders, and the inhibition of endosomal TLRs has been considered as a possible therapeutic strategy in these clinical contexts [22,23,24,25,26,27].

Naturally, the development of potent and specific small-molecule TLR7 antagonists as therapeutic agents is of significant interest, although none is currently available for clinical use. The antimalarial agent hydroxychloroquine (HCQ), which operates as a nonselective endosomal TLR inhibitor, is widely utilized in the treatment of various autoimmune disorders [28]. Various researchers have reported molecules with quinazoline [29,30], benzoxazole [31,32,33], imidazopyridine [34,35], purine [36,37,38], and other varied scaffolds [39,40] that have been significantly investigated for the development of selective small-molecule TLR7 antagonists in several recent studies through empirical screenings and activity-guided strategies.

In our previous structure-based drug design approach, we demonstrated a universal binding model hypothesis by exploiting a library of all the reported TLR7 antagonists from different chemotypes [41]. Based on our universal model, we displayed the role-specific substituents in TLR7 antagonism, thus, experimentally validating our model. The structure-based exploration revealed critical structural attributes and their importance in terms of engaging various pockets and grooves in the binding model. The goal of the present manuscript is to extend the exploration of ligand-based drug development through a QSAR model by correlating physicochemical properties with corresponding experimental activities. It involves the development of such models by systematically exploiting both 2D and 3D molecular features and also including a 3D-QSAR-based pharmacophore substructure. The validated models further pave the development of potent hit compounds by using the underlying insights derived from the models. We seek to investigate several substitutions at the C2, C4, and C7 positions on the quinazoline scaffold in a concerted way to affect TLR7 antagonism for the future development of novel compounds with potential therapeutic applications. We also evaluate the probable probe–receptor interactions of newly developed hit compounds by performing a molecular docking study against the TLR7 receptor. The docking study, utilizing our previously proposed binding model hypothesis, culminates in a satisfactory agreement between the structure-based and ligand-based approaches. We also evaluate the stability of the docked complexes with a subsequent molecular dynamics study. The manuscript reports in silico ADME, drug-likeness, and toxicities of the selected lead compounds that highlight the legitimacy of the whole approach of rational design of TLR7 antagonists and their target-binding in terms of physiological simulation.

## 2. Results

The important structural attributes responsible for the potent TLR7 antagonistic activities were determined through a ligand-based study including 2D-QSAR, 3D-QSAR, and pharmacophore modeling. The input dataset consisted of a diverse set of 54 small-molecule TLR7 antagonists having different chemotypes, along with their diverse biological activities, from the previously reported literature (Appendix A) [29,32,33,35,41]. All the experimental activity values were collected by following a similar biological assay procedure against the HEK293 reporter cell line.

### 2.1. 2D-QSAR Model

#### 2.1.1. Development of 2D-QSAR Model

The 2D-QSAR model, employing a dataset of 54 molecules split in a 70:30% ratio, was primarily developed to establish a quantitative correlation between the 2D structural features and the corresponding experimental activities. It was computed with a multiple linear regression (MLR) approach involving various 2D descriptors available within the AlvaDesc v2.0.4 tool via an online chemical database v4.2.131 (https://ochem.eu, accessed on 7 February 2022) [42]. The model, which satisfied the external and internal validation criteria, was subsequently chosen, and a linear relation between the activity (expressed as the negative logarithm of IC_50_ or pIC_50_) and the values of the five descriptors of the chosen model was established, represented as the following equation. The corresponding standardized coefficient of the descriptors is provided in Appendix A and represents their significance towards their activities.
pIC_50_ = −6.2155 + 0.1409 × VE3sign_D/Dt + 4.1832 × SpMin2_Bh(s) + 0.0366 × P_VSA_logP_5 − 0.9329 × Eig02_EA(dm) − 0.1016 × CATS2D_09_AA(1)

VE3sign_D/Dt: the logarithmic coefficient sum of the last eigenvector from the distance–detour matrix [43].SpMin2_Bh(s): the second-smallest eigenvalue of the Burden Matrix of the H-filled molecular graph weighted by intrinsic state [44].P_VSA_logP_5: the sum of Van der Waals surface area of atoms having a logP value in the range from 0 to 0.25 [43,45].Eig02_EA(dm): second eigenvalue from the edge adjacency matrix weighted by the dipole moment [43,46].CATS2d_09_AA: the number of hydrogen bond acceptors at an in-between topological distance of nine bonds [43,47].

The following are the statistical fitting parameters: coefficient of multiple determination (R^2^): 0.86; adjusted R^2^ (R^2^_adj_): 0.84; R^2^ - R^2^_adj_: 0.02; lack of fit (LOF): 0.17; root mean square error in training set prediction (RMSE_tr_): 0.30; mean absolute error on training set (MAE_tr_): 0.22; concordance correlation coefficient (CCC_tr_): 0.93; standard error of estimate (s): 0.33; and F-statistics value: 40.80.

Apart from the above five-descriptor model, several other models were also generated that could not attain either a satisfactorily high fit value, expressed as R^2^, or robustness, expressed as the cross-validation correlation coefficient using the leave-one-out method (Q^2^_LOO_), or that failed to achieve the criteria of the external validation parameters. The statistical significance of the selected model was supported by the high correlation coefficient between the experimental versus predicted activities with a value of 0.86 (Figure 1) and an almost equally high R^2^_adj_ value (0.84, difference: 0.02), which eliminated the scopes of over-fitting [48]. The error or deviation values, such as LOF (0.17), RMSE: 0.30, MAE (0.22), and the s value (0.33), were all below 0.5, indicating a satisfactory reliability of prediction [49].

In addition, the frequency of the descriptors comprised in the selected model was evaluated among all the generated five-descriptor models. All the top-occurring descriptors were found to be comprised in our selected model (Figure 2). This showed that the selected descriptors were chosen according to their consistent importance.

#### 2.1.2. Validation of 2D-QSAR Model

Apart from the fitting criteria explained above, which elucidated the quality of the correlation, additional cross-validation technique (internal validation) parameters such as variance in prediction using leave-one-out (Q^2^_LOO_: 0.83), leave-many-out (Q^2^_LMO_: 0.80), and Y-scramble (R^2^Y_scr_: 0.13; Q^2^Y_scr_: −0.28) methods were employed for better assessment of the precision and robustness of the model. The high average Q^2^_LMO_ value of the random models generated from 2000 iterations with larger perturbations was quite close to the R^2^ and Q^2^_LOO_ values, indicating the stability of the selected model [50] (Figure 3A). The doubt of chance correlation was eliminated, as demonstrated from the Y-scramble plot when the R^2^ and Q^2^ values of the selected model were far and much greater than the average R^2^Y_scr_ and Q^2^Y_scr_ values, respectively (Figure 3B) [51].

##### Internal Validation

The following values were achieved for internal validation: Q^2^_LOO_: 0.83; R^2^ − Q^2^_LOO_: 0.03; root mean square error for cross-validation (RMSE_cv_): 0.34; mean absolute error of cross-validation (MAE_cv_): 0.26; predictive residual sum of squares of cross-validation (PRESS_cv_): 4.31; concordance correlation coefficient (CCC_cv_): 0.91; Q^2^_LMO_: 0.80; R^2^Y_scr_: 0.13; Q^2^Y_scr_: −0.28; and root mean square error for scrambled predictions (RMSE_AV_Y_scr_): 0.77.

The external validation of the model showed a high squared correlation coefficient of the predicted and experimental activities (R^2^_ext_: 0.78). The predictive squared correlation coefficient (Q^2^_F1_) [52,53], which acted as the LOO cross-validation for the test set, along with other variance parameters for external predictions such as Q^2^_F2_ [54] and Q^2^_F3_ [55], all measured values greater than the threshold of 0.6 [56]. The validation parameters of CCC and R^2^m_aver_ also passed the criteria of being greater than 0.85 and 0.5, respectively [56].

##### External Validation

The following values were achieved for external validation: root mean square error for external prediction (RMSE_ext_): 0.37; mean absolute error for external prediction (MAE_ext_): 0.30; predictive residual sum of squares for external prediction (PRESS_ext_): 2.21; R^2^_ext_: 0.78; Q^2^_F1_: 0.71; Q^2^_F2_: 0.71; Q^2^_F3_: 0.80; CCC_ext_: 0.86; closeness between R^2^ and origin forcing R^2^ determination coefficient (R^2^m_aver_): 0.66; and closeness between R^2^ and origin forcing R^2^ (R^2^m_delta_): 0.15.

#### 2.1.3. Applicability Domain

The applicability domain is a region within a Williams Plot (scatter plot of standard residual response and leverage values) bounded by 2.5 standard deviation units in the ordinate and a HAT leverage threshold value in the abscissa side [57]. The 2.5σ limit was chosen to narrow the domain even more than 3σ, which covered 99% of the distributed data [58]. The applicability domain as expressed in the William Plot (Figure 4) indicated a threshold leverage value (h*) of 0.474, which encompassed a majority of the molecules within the domain, with leverage values lower than that, except for only three compounds of the training set as outliers (compounds **26**, **40**, and **44**). Even a stringent standardized residual threshold of 2.5σ managed to embrace all the molecules, except only a single outlier (compound **9**), indicating that the model could be considered for the design of new TLR7 antagonists with improved activities.

### 2.2. Pharmacophore Model

#### 2.2.1. Development of Pharmacophore Models

The entire molecule dataset randomly split into 70:30% (training: test set) resulted in 37 molecules in the training set and the remaining 17 molecules in the test set. Using the training set of 37 compounds from four activity sets, 10 pharmacophore models were created by combining the hydrogen bond acceptor (HBA), hydrophobic (HY), hydrophobic aromatic (HYA), positive ionizable (PI), and ring aromatic (RA) features, as recommended by feature mapping methodology. The top-ranked, best pharmacophore, Hypo1, was selected from the ten developed pharmacophores based on lowest total cost, highest cost difference, high correlation, and the low RMSD value exposed (Table 1). The ten developed pharmacophore hypotheses had total cost values ranging from 81.50 to 99.54 bits. One hydrogen bond acceptor (HBA) feature, one hydrophobic aromatic (HYA) feature, one positive ionizable (PI) feature, and one ring aromatic (RA) feature were the four pharmacophore features that built the basis of Hypo1 (Figure 5).

In the overview of all ten generated pharmacophore models, the corresponding correlation values between the experimental and predicted activity values were found to be quite consistent and higher than 0.9. This implied that each of them was individually quite reliable in predicting the activity values with their corresponding pharmacophore features. Among them, the significance of the top-ranked pharmacophore model, Hypo1, with a correlation coefficient of 0.94 and a RMSD of 0.87 (Table 1), was our model of choice for pharmacophore generation.

All the compounds were categorized into four different groups depending upon their experimental activities (IC_50_): most active (IC_50_ ≤ 2 μM, ++++), active (2 μM to 10 μM, +++), moderately active (10 μM to 20 μM, ++), and inactive- (IC_50_ > 20.0 μM, +). The predictive capability of the training set compounds is shown in Table 2. The activity levels of the most active compounds in the training set were all predicted extremely precisely, which demonstrated the predictive power of the Hypo1 model.

In the training set, compound **32** (IC_50_: 0.43 µM) and compound **14** (IC_50_: 0.7 µM) (Figure 6A,B), belonging to the most active set, overlapped with the pharmacophore hypothesis of Hypo1. The justification for this lies in the fact that they were satisfactorily mapped with all the significant features of the pharmacophore; however, in the case of the other molecules, some of the essential features were not mapped. Figure 6C depicts the mapping of Hypo1 with inactive compound **44** (IC_50_: 684 µM), which was not properly placed in the distantly located PI feature, and the HYA feature, despite being adjacent to the molecule, overlapped with a nonaromatic point of the molecule.

#### 2.2.2. Pharmacophore Validation

Three evaluation procedures, namely a cost analysis, a test set analysis, and Fischer’s randomization tests, were implemented for the validation of the best pharmacophore model (Hypo1).

##### Cost Analysis Method

In our study, the fixed cost and the null cost values of 10 generated hypotheses estimated with the Hypogen algorithm were 105.134 of 215.593 bits, respectively. A total cost value of 123.676 bits was observed for the selected Hypo1 model. Compared to the total cost values of the ten developed pharmacophore models, the first model possessed a value that was closer to the fixed value. The cost difference was computed with a distinction between the total cost and null cost that ought to be greater and be between the total cost and fixed cost values for a significant pharmacophore model. A cost difference of 40 to 60 indicated that the corresponding pharmacophore model was more than 90% reliable in correlating the experimental and predicted activity values. Among the 10 generated pharmacophore models, the highest cost difference value of 91.917 bits was observed for the Hypo1 model, along with the lowest root mean square deviation of 0.998167 Å (Table 1). Due to the high value of cost difference, Hypo1 was the most significant model for predicting the experimental activities (IC_50_) of the training set compounds with a precision of >95% statistical significance. 

##### Test Set Analysis

The degree of effectiveness of the selected pharmacophore model was dependent on the capability to estimate the biological activity of 17 test set compounds, along with the training set molecules. The same four orders of magnitude were applied against the test set molecules. To evaluate the significance of the pharmacophore model, Hypo1 was mapped with the bound conformation of the test set compounds, which unveiled the probable estimated activity of the test set compounds. Most of the molecules in the test set with diverse structural features were correctly predicted (Table 3). 

The ligand pharmacophore mapping provided one of the most active compounds, **31** (IC_50_ = 0.46 μM), from the benzoxazole core, which was highly correlated with all the essential features of the Hypo1 pharmacophore model (Appendix A). From the quinazoline core, compound **29** (IC_50_ = 1.83 μM) was also perfectly mapped with all the pharmacophoric features (Appendix A). However, the positive ionizable (PI) and HBA feature location vector maps were not able to map inactive compound **46** (IC_50_ = 660 μM) (Appendix A), which illustrated a satisfactory correlation between the features of the test set and their corresponding biological activities.

The estimated activities of both the training and test set compounds were correlated with their experimental activities using a regression analysis with the correlation values of 0.94 and 0.92, respectively (Figure 7).

##### Fischer Randomization

In this approach, 19 random, different spreadsheets were prepared while maintaining 95% statistical confidence with randomly scrambled training set activities by utilizing similar parameters to those used to prepare the actual pharmacophore model [59]. If the randomized generated hypothesis possessed similar or better statistical confidence, it indicated that the actual hypothesis was built in an unbiased manner [60,61]. High cost values of 19 individual spreadsheets were observed for the total cost of the Hypo1 model, and the correlation value was less than that of the Hypo1 model (Appendix A) (Figure 8). These results signified that Hypo1 was more statistically significant than other randomly generated pharmacophore models and that Hypo1 can be used as a validated model for developing a chemical library of TLR7 inhibitors with a variety of structural features.

### 2.3. 3D-QSAR

#### 2.3.1. Development of the 3D-QSAR Model

A robust 3D-QSAR model was created to attain a structure–activity relationship profile of 54 energy-minimized TLR7 antagonist compounds (Appendix A), as well as to evaluate the role of potential electrostatic and steric fields responsible for TLR7 antagonistic activity. To calculate the activities (pIC_50_) of the newly developed compounds, the best 3D-QSAR model was used. A good alignment of the 54 TLR7 inhibitors is essential for molecular field analysis in 3D-QSAR modelling. A molecular overlay tool was adopted to superimpose 54 inhibitors with energy-minimized conformations. In the 3D-QSAR analysis, the negative logarithm of the IC_50_ (pIC_50_) of those inhibitors was used as a dependent variable (Appendix A).

The training set and test set molecules were chosen randomly to reflect the variability in structure and activity throughout the whole dataset by considering that the test set molecules reflected a biological activity scale that was close to the biological activity of the training set. The whole group of compounds was split into a training set (37 compounds) and a test set (17 compounds) in a 70/30% ratio. Using the 37 aligned training set molecules, a 3D-QSAR model was established. The potential power of the developed 3D-QSAR model depended (Appendix A) on the ability to evaluate the pIC_50_ values of the training set.

A preliminary analysis was conducted to determine the relative significance of each field and its impact on the TLR7 antagonistic activity. The steric and electrostatic field contributions for individual compounds were determined as independent variables in the developed 3D-QSAR model. The subsequent studies were conducted with the potential steric and electrostatic fields measured simultaneously at each grid point to predict and interpret the TLR7 inhibitory activities of the different molecular scaffolds. A partial least square analysis was performed for all the individual training set and test set molecules to linearly correlate the experimental and estimated activities.

#### 2.3.2. 3D-QSAR Model Validation

Different combinations of electrostatic and Van der Waals (steric) fields were utilized to construct the various 3D-QSAR models for comparative molecular field analysis. The effectiveness of a QSAR model depends on its ability to accurately and reliably predict the activity of the newly designed compounds. The molecular fields were linearly correlated to the inhibitory activities using a partial least square approach. External validation of the 3D-QSAR model was achieved by predicting the activity of TLR7 inhibitors reported in the external test set (Appendix A).

The collected independent variables from the training set were subjected to a cross-validated PLS analysis against the external test set to determine the significance of Q^2^_test_ in the generated 3D-QSAR model (0.515 for two components), whereas the noncross-validated PLS model depicted a Q^2^_test_ value of 0.84. The R^2^_training_ correlation coefficient between the predicted and experimental activities of the training set was 0.95, indicating that the developed 3D-QSAR model was a significant model for investigating the molecular field effect of the 54 TLR7 inhibitors. Figure 9 indicates a strong correlation between the expected and experimental pIC_50_ values for both the test and training sets. As a result, this generated 3D-QSAR model was reliable and could be used to design new compounds and predict their potential activities.

#### 2.3.3. Analysis of 3D-QSAR Contour Maps

The biggest benefit of 3D-QSAR methodologies is that a contour map can be used to visualize the field effect on the target biological activities. Contour maps of 3D-QSAR models include information that can be applied to recognize significant areas in a 3-dimensional space surrounding the molecules, where changes in the steric Van der Waals isosurface (green and yellow grids) and electrostatic potential fields (blue and red grids) can have a direct impact on TLR7 inhibitory activity (Figure 10). The results obtained from the 3D-QSAR model served as a model for the development of new TLR7 inhibitory compounds. Overall, the present 3D-QSAR research investigated the important structural characteristics of several chemical classes of compounds that may be exploited to improve TLR7 inhibitory action by modifying the structural features of the lead molecules.

On the electrostatic contour map (Figure 10A), areas around the red contours show where high electron density was anticipated to improve the biological activity, while blue contours identify regions where low electron density was expected to increase the biological activity. The green isosurface marked on the steric contour maps (Figure 10B) denotes where bulky groups were preferred for an increase in activity, while yellow indicates a region where bulky groups were unfavorable. Among all the molecules of the entire dataset, those having quinazoline and benzoxazole cores widely showed comparatively more potent TLR7 antagonistic activities. Thus, the 3D-QSAR contour map interpretations were restricted to those molecular cores that correlated the 3D-QSAR molecular field with their activities for further structural exploration.

According to the molecular mappings, there were no sites for electrostatic potential grids or Van der Waals grids for ring A and ring B of quinazoline core template molecule **14** (Figure 11), implying that those locations were beyond the scope of any further modifications in this work. The 3D-QSAR isogrids revealed that electrostatic (red contour) and sterically (yellow) unfavorable groups enclosed the ortho and para positions of the piperazine moiety (C ring) of template molecule **14**, where less bulky substituents having high electron density were needed to increase the activity (Figure 11). Apart from that, the VDW and electrostatic grids confirmed that some electron-withdrawing, bulky substituents were more favorable at the position of a flexible, three-carbon linker attached at the C7 position. The ortho position of the attached pyrrolidine moiety (D ring) could be replaced with an electron-donating substituent due to the presence of a favorable electrostatic field around it. Similar to the structure-based study, a more bulky group attachment instead of a small dimethylamine group at the C2 position of the main quinazoline ring could be one of the probable possibilities to increase antagonistic activities.

Apart from field analysis on the quinazoline core, we also interpreted an important molecular field analysis of the benzoxazole scaffold based on template compound **32** (Figure 12). A negative coefficient on the electrostatic field, a positive steric contour map surrounding the aromatic benzyl group (D ring), and a flexible, three-carbon linker group (Figure 12) signified that bulky, electronegative groups at this position were favorable for increasing inhibitory activities. The 3D-QSAR contour map also showed a small, electrostatically favored, blue contour enclosing the dimethyl amine attached to the pyrrolidine group (A ring) of the template molecule (compound **32**), where more electron-donating substituents were expected to increase activity. Another large, electrostatically unfavored (red grid) but sterically favored contour (green grid) map (Figure 12) located near the nitrogen of the pyrrolidine group (A ring), along with the attached, flexible three-carbon linker of template compound **32**, suggested that more bulky, negatively charged groups at this position were favorable to increase the inhibitory activities. The terminal dimethylamine group attached with pyrrolidine (A ring) could be replaced with a bulky, electron-donating group to enhance activity.

### 2.4. Design of New Compounds

The cumulative ligand-based pharmacophore and 3D-QSAR study using all 54 reported molecules with diverse activities highlighted the importance of different electrostatic, steric, and hydrophobic feature distributions of the molecular dataset with significant correlation to their corresponding TLR7 inhibitory activities. Structural modification on the 3D-aligned quinazoline template molecule (compound **14**) was initiated considering the 3D-QSAR field effects and retaining the pharmacophoric features. Moreover, for any substitution, structural insights indicated from the 2D-QSAR were also considered. The 2D descriptor values of the designed quinazoline molecules were also found to follow the 2D-QSAR model equation (Appendix A). The designed molecules were also found to be located properly within the applicability domain of the 2D-QSAR model. This was evident from the Insubria plot (a variation of a Williams Plot to predict results for chemicals that have no experimental data available, replacing the standardized residual with the predicted value on the y-axis) [62,63], which established the validity of the designed molecules within the domain of the 2D-QSAR model (Figure 13).

Thus, we combined the steric and electrostatic field effects (derived from 3D-QSAR), as well as the hydrophobic features (from pharmacophore) of quinazoline core compound **14**, for the designing of new compounds with improved antagonistic activity, as shown in Figure 14.

We started by employing various substitutions at the C2 and C7 positions of the quinazoline ring, as this scaffold aligned with the pharmacophoric RA and HYA features (Figure 15 and Appendix A) and is important for hydrophobic *π*–*π* interactions with the Tyr356* and Phe408* (* represents the residues of chain B of the homodimeric protein) residues in the proposed binding model [41]. We retained the piperazine moiety as a useful surrogate at the C4 position of the quinazoline ring because the terminal nitrogen atom of piperazine, being prone to protonation, acted as a hydrogen bond donor. We installed various less bulky, electron-donating aromatic substituents on the piperazine moiety in compounds **T55**, **T56**, **T57,** and **T58** to reach the sterically unfavored positive electrostatic field around them (Figure 14), which resulted in a predicted IC_50_ < 2 µM (in both 3D-QSAR and Pharmacophore) (Table 4).

Since these TLRs are expressed in the acidic (pH 5.5–6.5) endosomal compartment, the molecules must access the target receptor protein located inside the endosomal compartment through their protonated state. We preserved the lipophilic, flexible, three-carbon linker at the C7 position to enhance the extent of the molecules across the hydrophobic pocket and the weak basic amine substituent, which in turn was protonated to engage the positive ionizable feature of the pharmacophore (Figure 15 and Appendix A) [29,41,64]. Thus, we first incorporated electron-donating amino groups on the propylpyrrolidine moiety with lysosomotropic properties, implicating a positive electrostatic potential field (blue grid) that led to compound **T63**, which probably exhibited significant predicted TLR7 inhibitory activities (Pred_IC_50_: 1 µM) (Table 4). Moreover, we incorporated more electropositive groups on the bulky, flexible propylpiperidine moiety with lysosomotropic effects at the C7 position to satisfy the surrounding low-electron-density contour map (blue grid) in compounds **T59**, **T60**, **T61**, and **T62** (Figure 14). They showed better antagonistic activities with predictive IC_50_ values of 1.57 µM, 1.25 µM, 0.94 µM, and 1.21 µM, respectively (Table 4).

### 2.5. Molecular Docking of the Newly Designed Compounds

A molecular docking study was performed to visualize the reason behind the TLR7 inhibitory activity of newly designed compounds with minor structural changes. We already built and validated a homology model structure of human TLR7 [41] and performed a binding analysis of those 12 compounds on their antagonistic binding sites using the Discovery Studio v18.1 client LibDock module. We developed a binding model of the diverse molecular library of TLR7 antagonist compounds with quinazoline, benzoxazole, chromenoimidazolone, and imidazopyridazine groups that could correlate minor structural modifications with corresponding inhibitory activities [41]. The electrostatic structure of the receptor–ligand interaction map revealed the significance of three hydrophobic pockets (pockets 1, 2, and 3) and two small grooves (grooves 1 and 2) on the binding site domain for TLR7 antagonistic activities (Figure 16) [41].

The docking analysis of all newly designed quinazoline compounds revealed that the quinazoline core was stabilized in the central cavity by establishing conventional hydrogen bond interactions between the core nitrogen, N1, and Gln354* (after protonation at pH 5.5–6.5) (Table 5). The quinazoline core in all the compounds, except compounds **T57**, **T58**, **T60**, and **T64**, also established a hydrophobic *π*–*π* stacking interaction with Tyr356* on the central cavity of the binding domain (Figure 17 and Appendix A). Compounds **T55**, **T56**, **T57**, and **T58** had an aromatic-substituted piperazine group at the C4 position that was orientated towards pocket 3, where the aromatic-substituted part protruded into groove 2 (Figure 17 and Appendix A). However, adding a bulky, aromatic substituent to the piperazine group in compounds **T55**, **T56**, **T57**, and **T58** prevented the piperazine nitrogen from protonation, which in turn made it unable to create a hydrogen bond with Thr525. Instead, the fluorine atom attached to the aromatic part in compounds **T55** and **T56**, making a conventional hydrogen bond interaction and halogen-bonding with Lys432* and Ser523, respectively (Table 5 and Figure 17), whereas the aromatic substituent attached to the piperazine moiety in compound **T57** participated in building a hydrophobic *π*–*π* stacking interaction with Phe500. 

The molecular docking analysis of the quinazoline molecules (compounds **T59, T60, T61**, **T62**, **T63**, **T64**, and **T65**) with a flexible, three-carbon linker at the C7 position showed a favorable binding conformation by orienting toward tunnel-shaped, hydrophobic pocket 1 (Figure 17 and Appendix A). The protonated nitrogen atom in the piperidine moiety attached to the flexible chain at the C7 position of compounds **T59**, **T62**, and **T64** served as a hydrogen bond donor to interact with Gly577 (Table 5) by, preferably, entering into hydrophobic pocket 1 (Figure 17 and Appendix A). However, in the case of molecules **T60**, **T61**, **T63**, and **T65**, the substituent group attached to the piperidine ring participated in hydrogen bond interactions with Gly577 after satisfactorily occupying pocket 1. However, in molecule **T65**, the substituent fluorine atom on the pyrrolidine ring formed a halogen bond with Thr406* (Table 5). The docking study indicated that the nature and type of substitution patterns presented on the designed quinazoline group of compounds were adequate to engage the distinct pockets and grooves explored and abided by the binding model proposed.

### 2.6. In Silico Pharmacokinetics Predictions

Dropouts of lead compounds during preclinical and clinical studies are frequently due to poor pharmacokinetic profiles and toxicity problems. It would be highly beneficial to the drug discovery process if these challenges could be traced early on. In light of these considerations, the use of in silico methods to predict ADMET characteristics is intended as a first step toward analyzing novel chemical entities to avoid wasting time on lead molecules that are toxic or metabolized by the body into an inactive form that cannot cross membranes [65,66]. Thus, the pharmacokinetic profiles of all 12 designed compounds were subjected to evaluation using six predeveloped and prevalidated ADMET models offered by the Discovery Studio v18.1 client program. The pharmacokinetic profiles of the designed compounds are summarized in Table 6 with a biplot (Figure 18). The biplot represents the two similar 95% and 99% confidence ellipses for the HIA and BBB models, respectively. The polar surface area (PSA) was shown to have an inverse relationship with the percentage of human intestinal absorption and cell membrane permeability [67].

The molecules with good absorption profiles were most likely to be found within the ellipses having confidence levels of 95% and 99% (Figure 18). The upper limit of the PSA_2D value for the 95% confidence ellipsoid was 131.62, while the upper limit of the PSA_2D value for the 99% confidence ellipsoid was 148.12. As per the model, the molecule should satisfy the criteria of (PSA < 140 Å^2^ and AlogP98 < 5) for optimal cell permeability [68]. Therefore, all the newly designed compounds here exhibited polar surface areas (PSAs) < 140 Å^2^, but some of the compounds also experienced higher AlogP98 values.

Compounds **T55** and **T59**, among all the others, showed comparatively greater logP values, indicating moderate intestinal absorption (level 1), and compound **T59** was also unable to penetrate the blood–brain barrier (Table 6 and Figure 18). The bioavailability of potential medicines is influenced by their water solubility. However, the majority of the compounds experienced moderate aqueous solubility levels, as referred to in Appendix A, with the exception of compounds **T55**, **T59**, **T64**, and **T66**, which had low aqueous solubility issues (calculated for water at 25 °C). One of the most significant enzymes involved in drug metabolism is CYP2D6. All the molecules except **T57** and **T58** were discovered to be non-inhibitors of cytochrome P450 2D6 (Table 6), implying that all the newly designed TLR7 inhibitors were well-metabolized in phase I metabolism. Furthermore, no hepatotoxicity was observed for any of the substances; thus, they were subjected to a substantial first-pass effect. Moreover, the drug-likeness properties of the proposed molecules are indicated their oral bioavailability in Table 7.

### 2.7. Toxicity Risk Assessment Screening

Apart from pharmacokinetic profiling, potent TLR7 antagonist compounds were further evaluated for their toxicity profiling using the DS_TOPKAT module. Various toxicity modules of the compounds are listed in Table 8. The toxicity risk assessment results showed that all the potent compounds were non-carcinogenic against both male and female mice (Table 8). Similarly, FDA carcinogenicity prediction was also conducted on male and female rats, where all the compounds exhibited their non-carcinogenic properties against both rats. Ames mutagenicity tests against all six potential hits exhibited their non-mutagenic behavior (Table 8). However, compounds **T59**, **T60**, **T61**, **T62**, **T63**, and **T65** showed mild skin irritancy. All the compounds were non-biodegradable (Table 8). This predictive drug toxicity profiling can guide the further development of potent TLR7 antagonists.

### 2.8. Molecular Dynamics Simulation

The protein–ligand complex system was simulated in Gromacs 5.1.5 on an Intel(R) Xeon(R) Silver 4214R CPU cluster to evaluate the stability of the ligand in the binding domain on the TLR7 homodimeric interface. All the atoms’ MD simulations were performed to further evaluate the stability of the three selected TLR7 antagonist molecules, **T55**, **T56**, and **T66,** with favorable pharmacokinetic profiles on their hydrophobic binding cavities. Their corresponding RMSD, RMSF, and radius of gyration (Rg) curves revealed that ligand-binding remained stable across the 10 ns trajectory. Cα atoms of the TLR7 protein backbone were fixed by fixing translational and rotational spinning to the corresponding initial structure for molecular dynamics runs during the RMSD calculations of the complex protein [69]. Upon the selected antagonists’ bindings to the homodimeric interface of the TLR7 ectodomain region, the RMSD curve deviated slightly (Figure 19A) for compound **T56**, resulting in conformational changes [41]. However, the RMSD patterns of both the native TLR7 protein and the inhibitor–protein complex remained highly consistent throughout the 10 ns MD simulation runs, indicating that the native TLR7 homodimeric protein did not change its backbone structure after the antagonist bindings (Figure 19A).

Additionally, the compactness of the protein and its change in folding behavior over the trajectories after the ligand binding was demonstrated by the radius of gyration (Rg) curves (Figure 19B). Throughout the simulation run, TLR7 antagonists **T55**, **T56**, and **T66** maintained a comparatively constant Rg value of approximately 3.87 nm, which represented the conserved interactions between the active residues and ligands that mediated the protein to fold more steadily. The ligand-free TLR7 protein had slightly higher Rg values (Figure 19B), which indicated that the protein backbone was less compact. The loop regions of protein backbones adopted the most stable folded conformations as the antagonists bound to them.

Moreover, the fluctuation of complex protein residues was compared to the unliganded native TLR7 protein, which acted as a reference structure, with a least square fitting method over a 10 ns trajectory run. The RMSF curve had two strong peaks on the first monomeric chains between the 440–450 residues for compound **T66** and the C-terminal regions of the second monomeric chain for compound **T56**, respectively (Figure 20). The fluctuations were mainly experienced in Z-loop regions due to proteolytic cleavage and in C-terminal loop regions due to cutting with the TIR domain regions. Compared to the native protein, the majority of the atoms in the complex structure had similar or lower fluctuations, indicating that the protein was more stable after ligand binding (Figure 20).

## 3. Materials and Methods

### 3.1. Dataset Selection

A broadly populated molecular dataset was selected with a library of 54 structurally diverse reported molecules (Figure 1) having TLR7 inhibitory activities against the HEK293 reporter cell line [29,32,33,35,41]. The compound dataset was arranged into four orders of magnitudes, named as most active, active, moderately active, and inactive classes of compounds. The most active set of compounds had IC_50_ values < 2 µM, followed by the active set with IC_50_ values from 2 µM to 10 µM, and the moderately active set with values from 10 µM to 20 µM, while the remaining molecules were kept in the inactive set.

### 3.2. 2D-QSAR

#### 3.2.1. 2D-QSAR Model Generation

Data on 54 TLR7 antagonist compounds, along with their biological activities, were collected from the previous literature [29,32,33,35,41]. Experimental activity (IC_50_) was uniformly maintained to the micromolar unit (µM) and converted to the logarithmic unit (pIC_50_) [70]. The molecules were subsequently aligned, energy-minimized through a consistent forcefield (CFF), and subjected to compute the 2D molecular descriptors using the AlvaDesc v2.0.4 tool via an online chemical database v4.2.131 (https://ochem.eu, accessed on 7 February 2022) [42] to eliminate the complexity of various 3D conformational spaces. In the beginning, 4369 descriptors were computed, including constitutional indices, topological indices, connectivity indices, 2D-matrix-based descriptors, edge adjacency indices, Burden eigenvalues, functional Group counts, P_VSA-like descriptors, pharmacophore descriptors, 2D atom pairs, molecular properties, etc., which were significant to describe the physicochemical properties and structural attributes of the dataset compounds. The descriptors having over-fitting biases and high correlations were excluded from the calculation according to the exclusion criteria of similarity > 80% and inter-correlation > 95% in QSARINS v2.2.4 [71,72]. After this progression, 639 descriptors were taken as selected descriptors for the advancement of the QSAR model. The 54-molecule dataset was divided into a training set (for model development) and a test set (for external validation) using a sorted response in QSARINS, where all the molecules were ordered by increasing activity. Apart from the most and least active compounds incorporated into the training set, two of every three remaining molecules were assigned to the training set and all others to the test set, keeping a ratio of 70:30%, respectively (Figure 21) [62].

The selected molecular descriptors were further subjected to a genetic algorithm-variable subset selection (GA-VSS) module [73] for selecting the significant subset variables with significance levels < 0.05 and 10,000 generations per size (up to 5 descriptors) for the robust QSAR model development. The selected subsets were adopted for the development of statistically significant and robust models with multiple linear regression (MLR) using an ordinary least square (OLS) model method [62,73] for all the training set data points in QSARINS v2.2.4 software.

#### 3.2.2. 2D-QSAR Model Validation

Following the OECD principles for QSAR validation [74], the developed models were first assessed for their fit values with a correlation coefficient (R^2^), adjusted R^2^ (R^2^_adj_), lack of fit (LOF), concordance correlation coefficient (CCC), and other measures of uncertainties (RMSE, MAE, etc.) Through internal validation, the robustness of the model was assessed using the leave-one-out (LOO) technique [75]. Additionally, with increased perturbation, the leave-many-out (LMO) technique was employed, which iteratively excluded 30% of the training compounds each time and generated random models that were close to the selected robust model [50]. To evaluate the true relation among the dataset dimensions, its structural heterogenicity, and the modelled response, a y-randomization test was performed where the biological activity values were shuffled randomly with 2000 scrambling iterations while the values of the descriptors were left unchanged, generating several random models [51]. The average squared correlation coefficient of the randomized model (R^2^Y_scr_) should, therefore, be smaller than the squared correlation coefficient (R^2^) of the selected model to nullify the doubt of chance correlation [76]. External validation was performed by applying the developed model equation to the training set. Apart from the general correlation coefficient (R^2^_ext_) and the measured Q^2^_F1_ [52,53], as proposed by OECD, other external validation parameters, such as Q^2^_F2_ and Q^2^_F3_ [55,77], were also measured. As mentioned by Chirico et al., these parameters could seem contradictory in cases, so additionally, a simpler criterion of CCC_ext_ (concordance correlation coefficient) was also considered [56].

#### 3.2.3. Interpretation of Descriptors of the Developed 2D-QSAR Model

VE3sign_D/Dt, the first descriptor of the 2D-QSAR model equation, is expressed as a negative coefficient. It represents *the logarithmic coefficient sum of the last eigenvector from the distance–detour matrix* [43] and is expressed as the following equation:
(2)$$\mathrm{VE}3\mathrm{sign}\_D/\mathrm{Dt}=\frac{n}{10}\log\left|\overset{n}{\underset{i=1}{\sum }}l_{i}\right|$$
where *l_i_* represents the coefficient of the eigenvector associated with the largest negative eigenvalue calculated on the distance–detour matrix [78]. The distance–detour matrix is a square symmetric matrix comprising the ratio between the shortest and the longest topological distances between two atoms in the constitutional molecular graph [79,80]. It is evident that this is a geometrical descriptor and does not draw any impact from the properties of the atoms. Thus, decreasing the number of detour or cyclic components (retaining at least some of them) in the molecular topology increased the value of this descriptor, which was favorable for activity. It was evident in compounds **36**, **38**, and **39**, which had high descriptor values, where the aliphatic straight chain was abundant, and the structure followed an overall linear connection.SpMin2_Bh(s) bears the largest coefficient value with a positive sign. It represents *the second-smallest eigenvalue of the Burden Matrix of the H-filled molecular graph weighted by intrinsic state* [43,81]. It is a square symmetric matrix expressed as:
(3)$${\left[B\left(\omega \right)\right]}_{ij}=\{\begin{array}{cc} \omega_{i},\; & if\;i=j\\ \sqrt{\pi_{b}}+0.001,\; & if\;i\;and\;j\;are\;connected\;and\;one\;is\;a\;teminal\;atom\\ \sqrt{\pi_{b}},\; & if\;i\;and\;j\;are\;connected\\ 0.001,\; & if\;i\;and\;j\;are\;not\;connected \end{array}$$
where *π_b_* is the bond order (1 for single, 2 for double, 3 for triple, and 1.5 for aromatic bonds) [43,82]; ω is the intrinsic value; an electrotopological index (I) of the atom is elucidated as I = $\frac{\left[{\left(2/N\right)}^{2}\delta v+1\right]}{\delta }$; *δv* and *δ* are the counts of valence and sigma electrons, respectively; and N is the principal quantum number [81]. Atoms in groups such as halogens, amines, and azide hydroxyl have comparatively high intrinsic values. Also, the number of unsaturation, especially connected terminal unsaturation, can contribute to the overall increase in the component values of the matrix. These aspects can have an impact on the descriptor being a positive contributor. Because the I-state was higher for electron-withdrawing groups (=N-: 3.00, >N-: 2.00, -O-: 3.50) [81], it was reflected in compound 37 with the highest descriptor value, which had several occurrences of such atoms.P_VSA_logP_5, the third parameter of the model, is a lipophilicity-based descriptor representing the *P_VSA-like on LogP, bin 5*, that is the sum of the Van der Waals surface area of atoms with logP values in the range of 0 to 0.25. This descriptor can also positively influence the activity, having a positively signed coefficient, and it is presented by both the size and hydrophobicity values of the atoms [45]. The Alvascience user manual [43] lists the individual octanol-water partition coefficient values of 115 atom-centered fragments, in which groups such as CR2X2, =CR2, =CX2, R:CR:R, R…O…R, and R-O-C=X specifically bear logP values ranging from 0 to 0.25 [83,84]. The atoms belonging to these and having larger atomic Van der Waals surface areas can be beneficially incorporated to enhance activity.
(4)$$VSA_{i}=4\pi R_{i}^{2}-\pi R_{i}\overset{nAT}{\underset{j=1}{\sum }}a_{ij}\left(\frac{R_{j}^{2}-{\left(R_{i}-d_{ij}\right)}^{2}}{d_{ij}}\right)$$
where *R_i_* is the atomic Van der Waals radius of the atom *i*, *nAT* is the number of atoms, *a_ij_* are the elements of the adjacency matrix, and *d_ij_* = min{max{|*R_i_* − *R_j_*|, *b_ij_*}, (*R_i_* + *R_j_*). In addition, *b_ij_* is the bond length between *i* and *j* (*b_ij_* = *r_ij_* − *c_ij_*); *r_ij_* is the reference bond length; and *c_ij_* is 0, 0.1, 0.2, and 0.3 for single, aromatic, double, and triple bonds, respectively [43].Eig02_EA(dm), or the *second eigenvalue from the edge adjacency matrix weighted by the dipole moment* [43,46], is a negative contributor with a significant coefficient value, which emphasizes that adjacent bonds with large dipole moments are likely to decrease activity. It indicates that the substituent groups having greater charge distributions inflicted by electronegative atoms can have a negative influence if the involved bond is branched and connected to several other components in the H-depleted molecular connection map.CATS2D_09_AA is *the number of hydrogen bond acceptors at an in-between topological distance of 9 bonds* [43,47] and points out the frequency of such occurrences as a negative contributor to activity. Although the central core bears several nitrogen atoms that can be potential hydrogen bond acceptors, a topological distance of 9 bonds is not very frequent. However, as in the case of molecule **3**, the symmetric pattern of the carbonyl oxygen atoms, piperazine ring, and fused pyrimidine contributed to the large value of this descriptor.

#### 3.2.4. Applicability Domain

To identify the region of chemical space where the QSAR could effectively predict the new compounds, the applicability domain was calculated [57]. The leverage threshold, h*, was determined using the following equation: h* = 3 × (k + 1)/n) [85,86], where n is the number of compounds in the training set, and k is the number of selected descriptors.

### 3.3. Pharmacophore Model Generation

All the molecular modelling and the 3D-QSAR studies discussed here were carried out on Intel Xeon workstations with the Discovery Studio v18.1 client molecular modelling program [87]. The 54 reported compounds were divided into a training set and a test set in a 70:30 ratio, respectively. For construction and validation of the pharmacophore model, a training set was formulated with 37 antagonist molecules with experimental activities ranging from 0.43 μM to 684 μM. The remaining 17 molecules with a similar activity spectrum (IC_50_: 0.46 μM to 660 μM) constituted the test set for validation purposes. Both the training and test sets were equally diverse and contained almost the same percentage of molecules belonging to different predefined activity categories (Figure 22). The same biological assays and assessment protocols were followed for the collection of the experimental activities of the antagonist molecules. The biological activity data for all compounds were collected after evaluation against a single cell line with the same bioassay condition. For all the individual training set molecules, a maximum of 255 different conformational positions was generated using the CAESAR conformational method within an energy cut-off of 20 kcal/mol above the global energy minimum [65,66,88].

#### 3.3.1. Generation of Pharmacophore Hypothesis with 3D-QSAR Pharmacophore Generation (Hypogen)

3D-QSAR pharmacophore methodologies were employed using the Hypogen algorithm, which could correlate the important chemical features present among the active molecules. The uncertainty value for all the molecules was set at 2 for both the training and test sets, which meant that the biological activity could be two times higher or lower than the actual values [89]. The roles of different chemical features in imparting differential antagonistic activities for the training set molecules were identified using feature mapping protocols in Biovia Discovery Studio v18.1 client [87]. The distribution of these significant chemical features on important training set compounds was determined with the 3D-fingerprints protocol. These features included the hydrogen bond acceptor (HBA), hydrophobicity (HY), hydrophobic aromaticity (HYA), positive ionizable (PI), and ring aromaticity (RA) features, which were considered important, and the corresponding experimental IC_50_ was referred to as an active property [90]. Minimum interfeature distances were restricted to 2 Å [91,92]. The least root mean square deviation (RMSD) value, the highest correlation value, and the lowest total cost were set as three significant parameters for the selection of the final pharmacophore from the 10 different generated hypotheses.

#### 3.3.2. Pharmacophore Validation

The resultant pharmacophore was validated to evaluate the quality of the generated model by using cost analysis [59,93], test set analysis, and Fischer’s randomization [59,61]. Fixed cost, total cost, and null cost, which were the three significant cost parameters calculated in the bits unit, dictated the quality of the model [94]. Null cost indicated that there was no correlation between the representative features and experimental data, which approximated the activity of the training set as average. Fixed cost reflected the cost of a simple, ideal model that could predict all the data perfectly. The total cost of the individual hypothesis was summarized over error cost, weight cost, and configuration cost [92]. The total cost should be close to the fixed cost and more distant from the null cost to develop a significant pharmacophore model. If the cost difference values between null cost and total cost were greater than 60 bits, there was a high probability of true correlation between the experimental and predictive activities. If the cost difference was less than a 40 to 60 bit range, the model should have a 70–90% predictability range [95]. In the test set validation method, the selected pharmacophore hypothesis was investigated to predict the biological activities of the previously segregated 17 test set molecules to assess how close the predicted antagonistic activity values were to the experimentally validated biological activity values. The generated pharmacophore estimated the activities of the test set compounds by mapping the ligand with the pharmacophore.

### 3.4. 3D-QSAR Model Generation

The 3D-QSAR methodology built regression models by utilizing whole molecular steric and electrostatic potential grids as an independent field for predicting activity and for visualizing favorable and unfavorable interactions.

#### 3.4.1. Molecular Alignment

The resulting 3D-QSAR model was always responsive to the specific alignment scheme, where structural alignment was probably the most subjective but crucial phase in the 3D-QSAR analysis [96]. The lowest energy conformation (minimized through CFF) of the most active compound (**32**, IC_50_: 0.43 μM) is usually used as a reference. The dataset was aligned using the common core of the active molecule as a reference template by using a field fit method based on the combination of steric and electrostatic fields available in Discovery Studio v18.1 client [70]. The alignment had a major impact on the prediction accuracy and statistical efficiency of the 3D-QSAR models. The proposed alignment and common substructure are depicted in Figure 23 [97].

#### 3.4.2. 3D-QSAR Model Development and Validation

The “random method” of the “Generate Training and Test Data” protocol in Discovery Studio v18.1 client was used to develop the training set and test set compounds. Seventy percent (37 compounds) of the compounds were utilized as a training set to construct the 3D-QSAR model, and the remaining thirty percent (17 compounds) was used as an external test set (Figure 24) to cross-validate the predictive potential of the developed 3D-QSAR model. The inhibitory activity of the compounds, IC_50_ (μmol/L), was first converted to a negative logarithmic value [pIC_50_ (μmol/L)], which was then used as the dependent variable in the 3D-QSAR study. A CFF was implemented on individual molecules. In Discovery Studio v18.1 client, the Van der Waals potential and the electrostatic potential were treated as separate individual descriptors for the building of two separate 3D-QSAR models. When the dielectric constant was compared to distance to simulate the influence of solvent, a positive point charge was employed as the electrostatic potential probe. The Van der Waals potential was measured using a carbon atom with a radius of 1.73 Å as a probe [98]. The 3D-QSAR models were created using the “Create 3D QSAR Model” protocol in Discovery Studio v18.1 client, and they used energy grids as signifiers to construct a partial least squares model. The grid spacing was normally fixed at 1.4 Å. The bounding box of all the ligands, plus a few angstroms of extension, was used as the extent of the grid. CFF was then used to quantify the energy potentials at each grid position. The generated model was validated by using a leave-one-out (LOO) approach in the cross-validation study [99,100] for internal validation and by employing the test set for external validation.

### 3.5. Molecular Docking

In Discovery Studio v18.1 client, the LibDock package was used to perform molecular docking studies of the newly designed potent TLR7 antagonist compounds on the binding site domain, maintaining an RMSD cut-off of 0.25Å. After removing all additives from the protein, hydrogen atoms were introduced by maintaining a CHARMM force field. A pH of 5.5 was used to replicate the endosomal protonation environment. The homodimeric hTLR7protein has two active site domains: one for small-molecule binding (binding site 1) and the other for ssRNA binding (binding site 2) [12,41]. These are found to be present at the intersection of two dimerized TLR7chains [101]. For the docking calculations, binding site 1 was used, comprising a sphere with a 9.5 Å radius, centering the binding area and encompassing all the critical regions for binding residues. A CAESAR conformational approach was used to dock the energy-minimized ligands, which allowed for a maximum of 255 conformations per compound within a 20-kcal/mol energy range above the global energy minimum threshold. During the docking program, a genetic algorithm was used to encrypt information about hydrogen bonding and hydrophobic *π*–*π* interactions into the binding sites of the hTLR7 homodimeric cavity, along with their plausible ligand-binding conformations.

### 3.6. ADMET and Toxicity Prediction

All twelve newly designed TLR7 antagonist compounds were employed for the estimation of their absorption, distribution, metabolism, elimination, and toxicity (ADMET) properties in Discovery Studio v18.1 client [87] to exclude compounds with unfavorable ADMET properties as soon as possible. The ADMET module utilized six mathematical models to quantitatively estimate the pharmacokinetic properties using a set of rules and keys (Appendix A) [65] that defined the threshold of the ADMET characteristics for the given quinazoline molecules. These models included human intestinal absorption (HIA), aqueous solubility, blood–brain barrier penetration (BBB), cytochrome P450 2D6 inhibition, and hepatotoxicity.

The absorption levels of the HIA model were specified by 95% and 99% confidence ellipses between the ADMET PSA 2D and ADMET AlogP98 surfaces [68]. The water solubility of each molecule at 25 °C was estimated using the ADMET aqueous solubility model [102]. This BBB model was derived from a quantitative linear regression model for the prediction of blood–brain penetration after oral administration, as well as 95% and 99% confidence ellipses in the ADMET_PSA_2D and ADMET_AlogP98 plane [103]. CYP2D6 predictions of the designed TLR7 antagonist compounds indicated whether the given molecules were capable of inhibiting the cytochrome P450 2D6 enzyme, which constitutes a drug–drug interaction [104]. The potential hepatotoxicity of the wide range of structurally diverse compounds was evaluated using ADMET hepatotoxicity.

Thereafter, the designed TLR7 antagonist molecules were subjected to various toxicity screening models, e.g., toxicity for carcinogenicity, developmental toxicity, mutagenicity, and skin irritancy or sensitization, using the DS_TOPKAT module of Discovery Studio v18.1 client [87]. These estimations assisted in the optimization of therapeutic ratios of the lead compound for the further development and analysis of any possible safety concerns.

### 3.7. MD Simulation

MD simulation of the docked conformations of the potent antagonists in the complex with the TLR7 protein homology model was performed using GROMACS 5.1.5 software on an Intel(R) Xeon(R) Silver 4214R CPU cluster [105,106]. The subsequent protein–ligand complex system was prepared as an initial structure using the pdb2gmx module in Gromacs. A Gromacs AMBER99SB force field [107] was applied for parameterization and topology generation of the native TLR7 protein. An AnteChamber Python Parser interface (ACPYPE) with GAFF parameters was applied to parameterize the required topologies, atomic types, and charges of ligands due to the presence of heteroatoms [94,108]. The complex was put in the center of a cubic box (9.3 × 9.3 × 9.3 nm^3^) filled with 285516 SPC/E water molecules to solvate the entire system [109,110]. Under physiological conditions (NaCl 0.15 M), the genion module in Gromacs was used to neutralize the whole system with a salt ion environment, followed by preserving electrical neutrality for the entire system. Finally, 27 Cl^−^ ions were incorporated into the system by replacing the water molecules to neutralize the whole system.

Energy minimization was used for 0.1 ns with a maximum force of 10.0 kJ/mol [111] to achieve the stable-state of the simulation system by eliminating discrepancies in atomic position or structural disputes, such as bond length and bond angle, as well as structural clashes between the positions of water molecules, ions, and protein complexes [109]. The energy minimization curve (Appendix A) revealed the quality of the energy-minimized structure. The equilibration of the protein–ligand complex system was carried out in two steps of a sequential process: (a) employing isothermal and isochoric ensemble (NVT) programs at a constant 300 K temperature and (b) using the NPT ensemble system at a stabilized zero bar pressure. Both ensembles executed 50,000 steps, which was equivalent to 0.1 ns. Finally, the temperature- and pressure-stabilized complex system was released for position restraining, where the solvent molecules in the cubic box were dissolved fully with the protein–ligand complex system. Subsequently, a position-restrained complex system was deposited for a 10 ns production MD simulation run in the presence of 310 K (V-rescale thermostat) temperature and atmospheric NPT ensemble pressure (Parrinello–Rahman barostat) and periodic boundary conditions with integrator time steps of 0.002 ps utilizing leap-frog algorithms. The LINC technique was used throughout the entire equilibration phase to restrict all hydrogen bonds [112,113], while a particle mesh Ewald (PME) module with a Fourier grid spacing of 0.16 was used to calculate long-range ionic interactions [114]. The whole trajectories were collected at a 2 fs timestep rate during the simulation for further investigation.

## 4. Conclusions

In the present work, we identified the detailed structural features present in the reported TLR7 antagonist compounds responsible for imparting their antagonistic activities, as well as the specific binding pattern against the endosomal TLR7 target protein. To identify significant chemical features responsible for TLR7 antagonism for further drug development, we intended to focus on developing ligand-based quantitative 2D and 3D-QSAR models and pharmacophore models. The top-ranked, best pharmacophore, Hypo1, was selected from ten developed pharmacophores. The 2D-QSAR study of the reported compounds presented the correlation of simple 2D physico-chemical descriptors with their inhibitory activities against TLR7 protein. This robust study involved 54 training set molecules based on an MLR algorithm with 2D descriptors selected using a genetic algorithm (GA) approach. Standard statistical metrics were used to validate the proposed model, both internally and externally. The generated 2D-QSAR model equation (R^2^= 0.8644) showed that the model was robust and effective and could be used to describe the TLR7 antagonistic activity of a wide range of compounds. The standard coefficient of the MLR equation reflected the significance of the edge-adjacency-based descriptor, Eig02_EA (dm), and pharmacophore descriptorCATS2D_09_AA negatively influencing the potent activity of the compounds.

Apart from that, a ligand-based quantitative 3D-QSAR pharmacophore model was developed based on 37 training set compounds with high diversity in terms of both chemical structures and biological activities. The significant chemical features responsible for TLR7 antagonism were identified using the best pharmacophore model, Hypo1, which was selected based on various parameters, such as cost difference, correlation coefficient, and other validation results. Hypo1 was created with one HBA feature, two positive ionizable (PI) features, and one hydrophobic aromatic (HYA) feature, with a cost difference of 91.917 bits. The predictive potential of the Hypo1 model was assessed using 17 test set molecules with a resulting correlation between predicted and biological activities of 0.9154 for the test set compounds, while for the training set compounds, the correlation was 0.9417. Apart from the pharmacophore model, another ligand-based 3D-QSAR study on a series of reported TLR7 small-molecule inhibitors yielded robust and statistically relevant predictive models, as demonstrated by moderate to high cross-correlation coefficients. Thus, the internal and external predictive powers of the obtained 3D-QSAR models were evaluated concerning the corresponding correlation coefficients (R^2^) between the estimated and experimental activities of the training set and test set compounds, which were 0.956 and 0.838, respectively. The developed contour maps of the 3D-QSAR models accurately indicated the contribution of combined electrostatic and steric molecular fields on corresponding TLR7 antagonistic activities.

The information-guided collective ligand-based drug design strategy including 2D-QSAR, 3D-QSAR, and pharmacophore models was utilized for the designing of novel TLR7 antagonist molecules. The combination of hydrophobic and hydrogen bond acceptor features, along with electrostatic and steric molecular field analyses, was used as a ligand-based strategy for the further designing of twelve new antagonists (**T55**–**T66**) by modifying the best quinazoline scaffold-based active template compound, **14**. Structural insights obtained from the 2D-QSAR model were taken into account for every substitution upon compound **14**. The activities of all twelve antagonist compounds were predicted through the 2D and 3D-QSAR models and pharmacophore model approaches.

A structure-based molecular docking study of the newly designed quinazoline scaffold compounds showed desired binding into the proposed hydrophobic pockets (pocket 1 and pocket 3), grooves (groove 1 and groove 2), and central cavity by maintaining hydrogen bond interactions with essential residues Thr525, Gly577, and Gln354*. The binding stability of the selected three compounds was evaluated with a 10ns MD simulation run. The RMSD, RMSF, and radius of gyration (Rg) curves revealed the status of the ligand binding inside the TLR7 protein. In silico ADMET assessments and drug-likeness properties of the newly designed compounds represented those that had similar kinetic properties compared to template compound **14**. The combined ligand-based drug design approaches, including 2D and 3D-QSAR studies and pharmacophore models, led to the design of new quinazoline scaffold compounds with potent TLR7 antagonistic activities. The ligand–receptor interaction model proposed in the manuscript can be used as a benchmark for the development of a new generation of potent TLR7 antagonists of clinical significance in various autoimmune diseases.

## Figures and Tables

**Figure 1 molecules-27-04026-f001:**
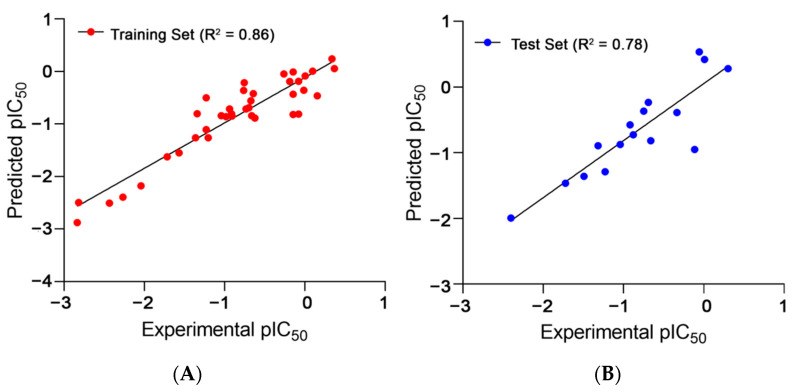
Observed and predicted TLR7 antagonistic activities of (**A**) training set and (**B**) test set compounds for the developed 2D-QSAR model.

**Figure 2 molecules-27-04026-f002:**
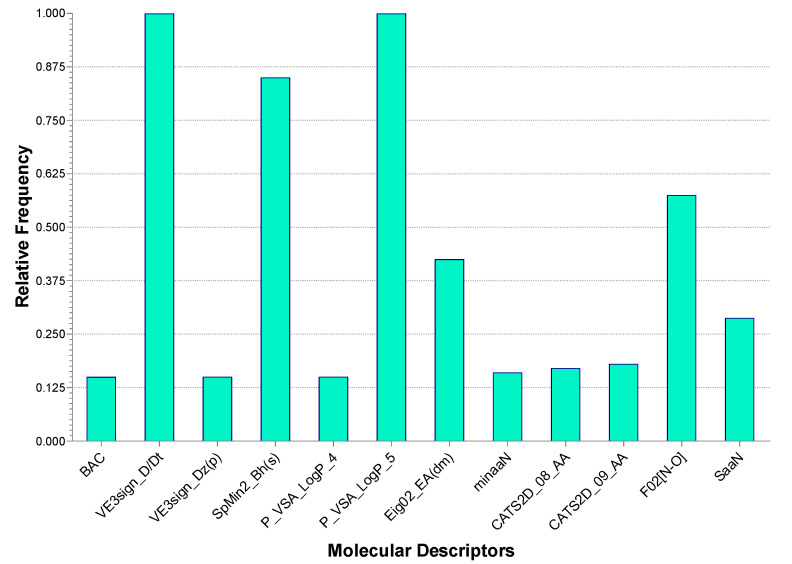
Occurrence of descriptors in the generated 5-descriptor models. BAC: Balaban centric index; VE3sign_Dz(p): logarithmic coefficient sum of the last eigenvector from Barysz matrix weighted by polarizability; P_VSA_logP_4: the sum of Van der Waals surface area of atoms having a logP value in the range from −0.25 to 0; minaaN: minimum E-state value of aromatic N; CATS2D_08_AA: number of hydrogen bond acceptors at in-between topological distance of 8 bonds; SaaN: sum of E-states of aromatic N.

**Figure 3 molecules-27-04026-f003:**
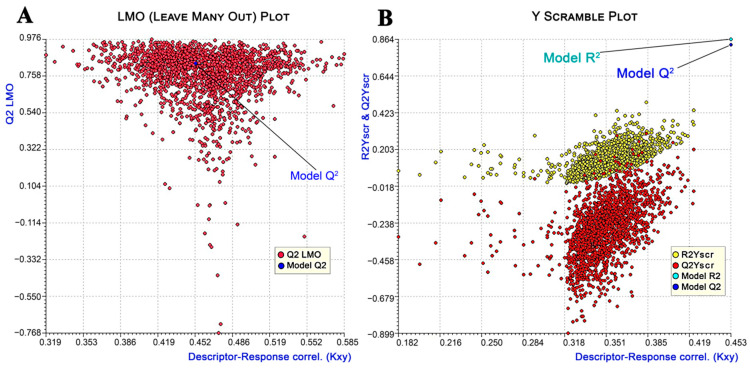
Leave-many-out (LMO) plot (**A**) and Y-randomization plot (**B**) of 2000 iterations for internal validation of the developed model.

**Figure 4 molecules-27-04026-f004:**
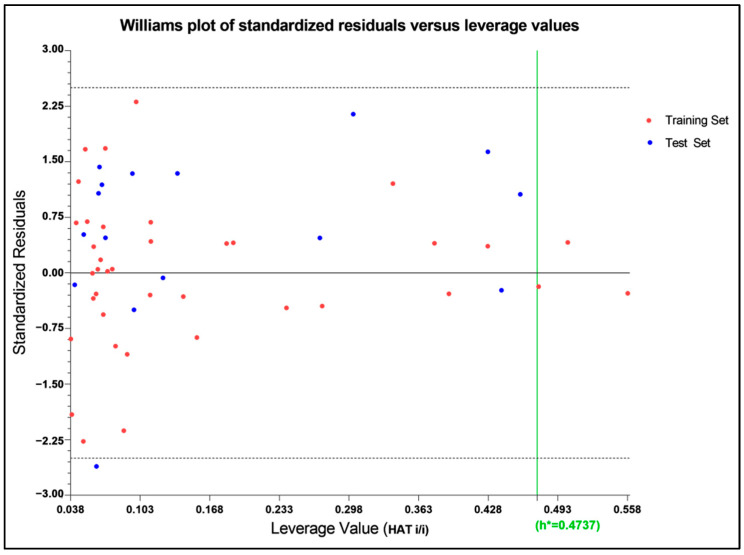
Willams plot to assess the applicability domain of the developed model. The threshold leverage value (h*) of the model is indicated by the green line which signified the majority of the molecules within the acceptable domain.

**Figure 5 molecules-27-04026-f005:**
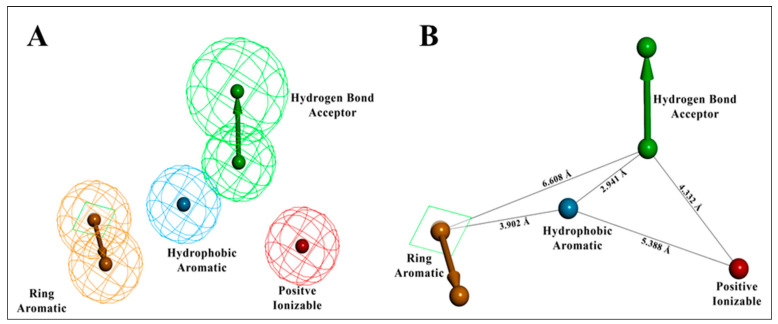
The best HypoGen pharmacophore model: Hypo1. (**A**) Chemical features present in Hypo1; (**B**) 3D spatial arrangement and the distance constraints between the chemical features. The green color represents HBA, the red color indicates the positive ionizable (PI) feature, the blue color represents the hydrophobic aromatic (HYA) feature, and the orange centroid represents the ring aromaticity (RA).

**Figure 6 molecules-27-04026-f006:**
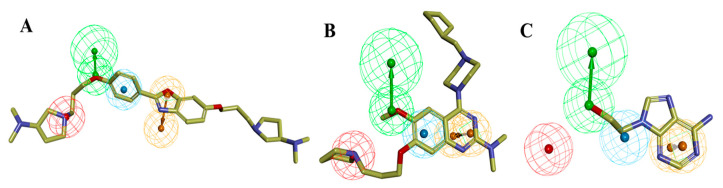
Ligand pharmacophore mapping of training set: (**A**) most active compound **32** (IC_50_: 0.43 µM), (**B**) most active compound **14** (IC_50_: 0.7 µM), and (**C**) inactive compound **44** (IC_50_: 684 µM). The pharmacophoric features HBA, HYA, PI, and RA are signified with green, blue, red, and orange, respectively.

**Figure 7 molecules-27-04026-f007:**
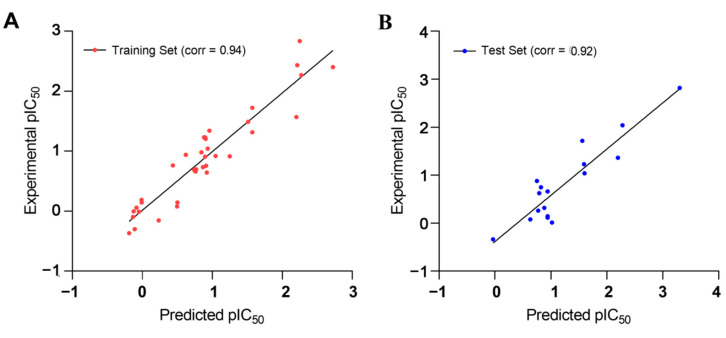
Observed and predicted activities of (**A**) training set and (**B**) test set compounds for the developed pharmacophore model.

**Figure 8 molecules-27-04026-f008:**
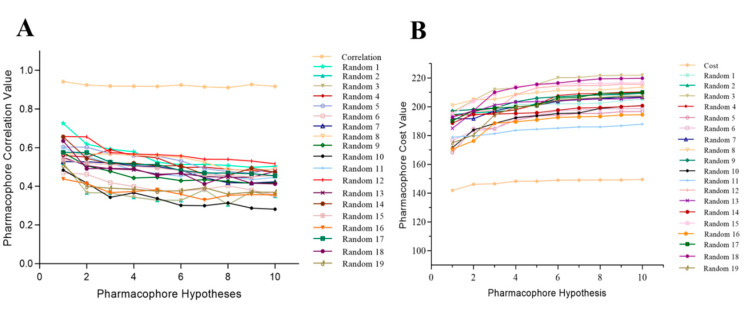
Difference in correlation (**A**) and cost values (**B**) of hypotheses between the selected pharmacophore models: Hypo1 and 19 random spreadsheets.

**Figure 9 molecules-27-04026-f009:**
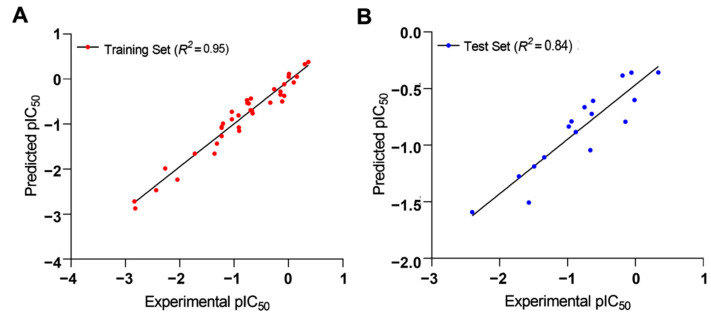
Plots of experimental TLR7 inhibitory activities versus predicted activities of (**A**) training set and (**B**) test set molecules for the 3D-QSAR model.

**Figure 10 molecules-27-04026-f010:**
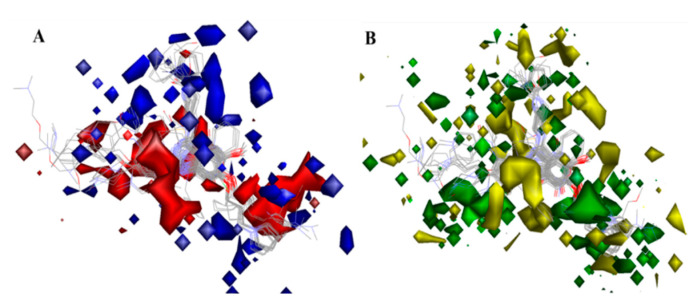
(**A**) 3D-QSAR model coefficients on electrostatic potential grids. Blue represents positive coefficients; red represents negative coefficients. (**B**) 3D-QSAR model coefficients on Van der Waals grids. Green represents positive coefficients; yellow represents negative coefficients.

**Figure 11 molecules-27-04026-f011:**
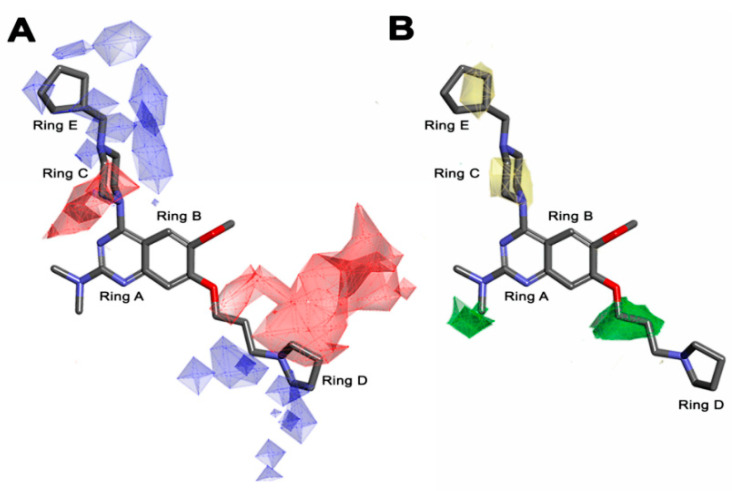
Pictorial representation of the 3D-QSAR model coefficients on (**A**) electrostatic and (**B**) Van der Waals grids mapping quinazoline core template molecule **14**.

**Figure 12 molecules-27-04026-f012:**
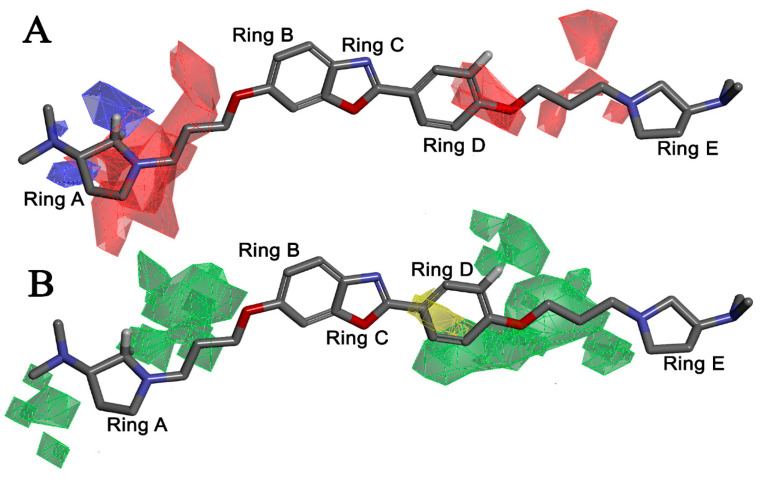
Pictorial representation of the 3D-QSAR model coefficients on (**A**) electrostatic potential and (**B**) Van der Waals (VDW) grid mapping benzoxazole core template molecule **32**.

**Figure 13 molecules-27-04026-f013:**
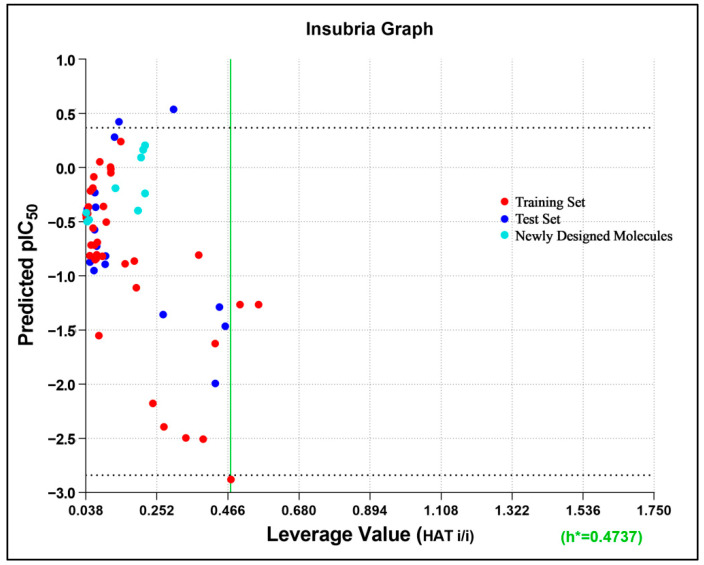
Insubria plot (plot of leverage hat values vs. predicted activity) of all data points. h* represents the threshold leverage value.

**Figure 14 molecules-27-04026-f014:**
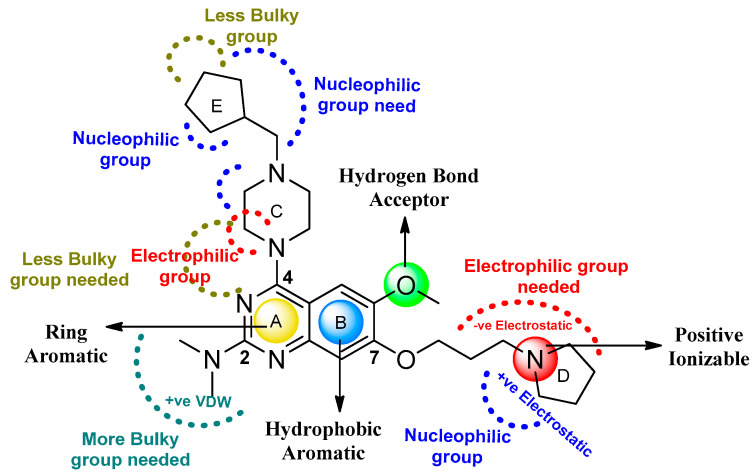
Distribution of structural attributes on quinazoline scaffold of compound **14** obtained from ligand-based drug design.

**Figure 15 molecules-27-04026-f015:**
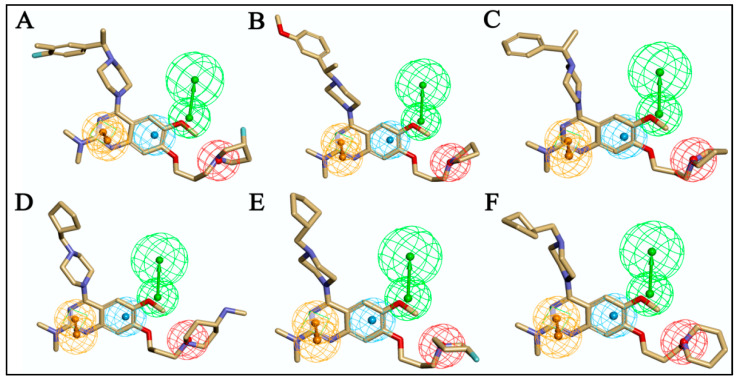
Alignment of six representatives of newly designed molecules on pharmacophore Hypo1: (**A**) **T55**, (**B**) **T57**, (**C**) **T58**, (**D**) **T60**, (**E**) **T62**, and (**F**) **T64**.

**Figure 16 molecules-27-04026-f016:**
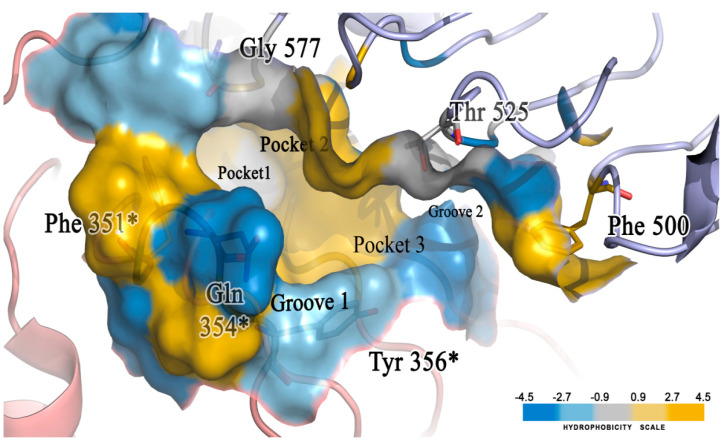
The binding pocket of TLR7 protein. The blue color represents low hydrophobicity, and the yellow areas represent highly hydrophobic regions with intermediates indicated in grey. * represents the residues of chain B of the homodimeric protein.

**Figure 17 molecules-27-04026-f017:**
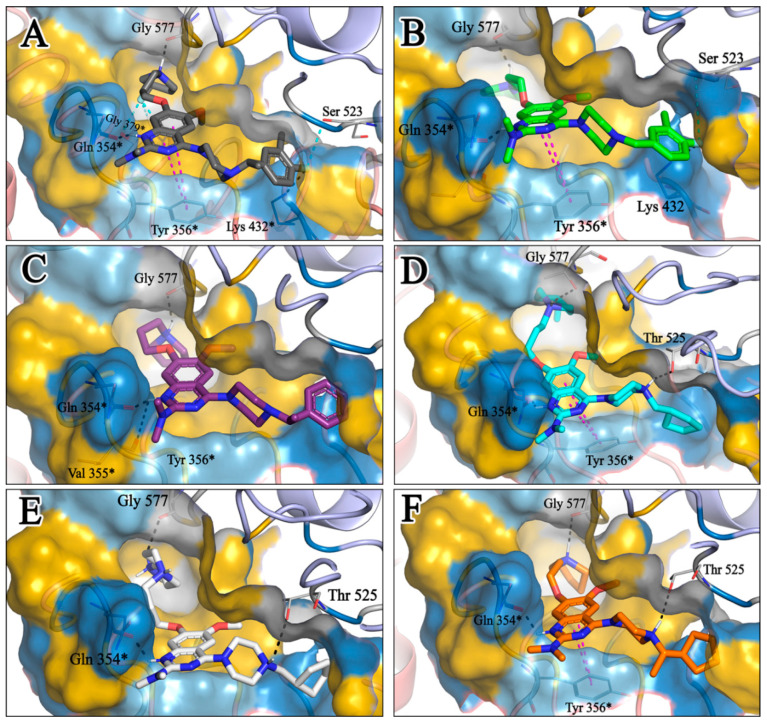
(**A**–**F**). Binding poses of compounds **T60**, **T56**, **T55**, **T66**, **T59**, and **T58**, respectively, into the proposed active site. Hydrogen bonds are indicated with black dotted lines, whereas purple dotted lines mean *π*–*π* hydrophobic interactions, and cyan dotted lines indicate halogen bonds.

**Figure 18 molecules-27-04026-f018:**
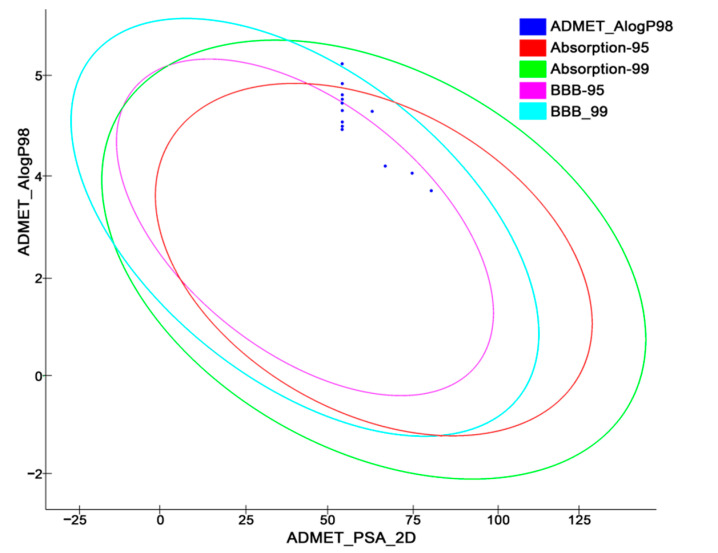
ADMET biplot curve showing ellipses having 95% and 99% confidence limits corresponding to the blood–brain barrier and intestinal absorption models.

**Figure 19 molecules-27-04026-f019:**
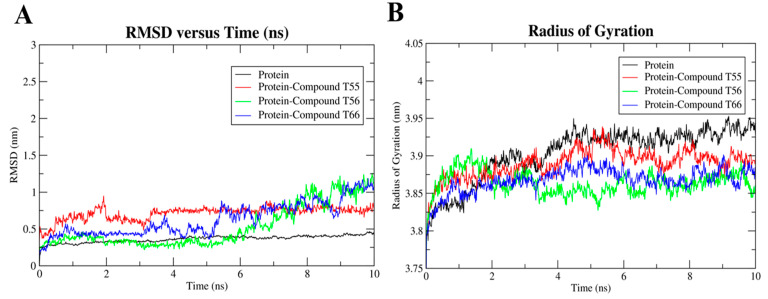
(**A**) Root mean square deviation plot and (**B**) radius of gyration plot of native TLR7 protein and protein–ligand complex over the 10 ns simulation.

**Figure 20 molecules-27-04026-f020:**
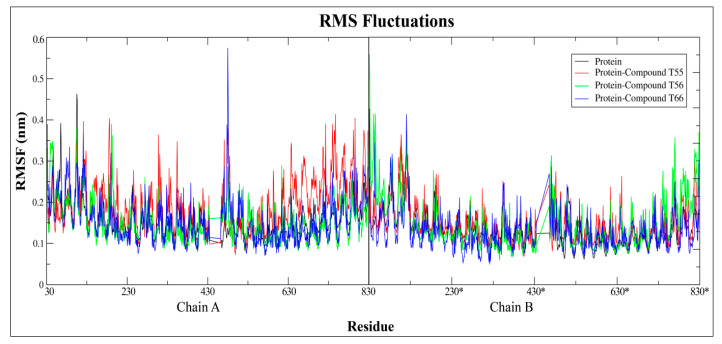
Root mean square fluctuations plot of native TLR7 protein and protein–ligand complex over 10 ns simulation run. * represents the residues of chain B of the homodimeric protein.

**Figure 21 molecules-27-04026-f021:**
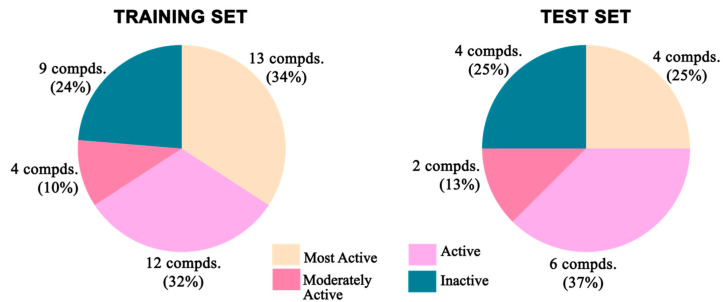
Distribution of different categories of compounds in training and test sets for 2D-QSAR study.

**Figure 22 molecules-27-04026-f022:**
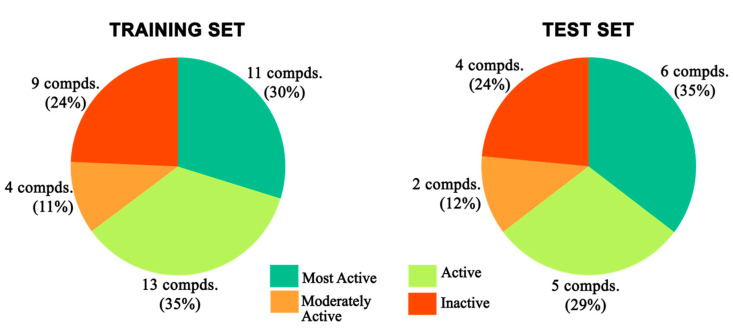
Distribution of different categories of compounds in training and test sets for pharmacophore modeling.

**Figure 23 molecules-27-04026-f023:**
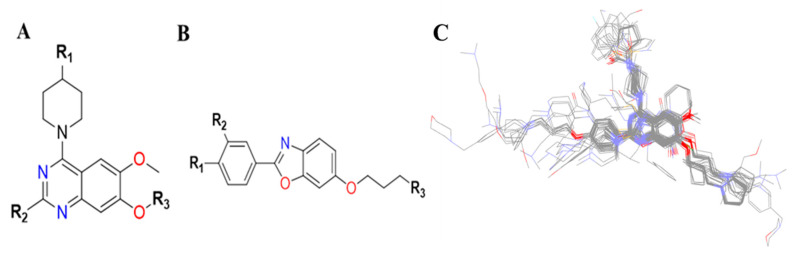
(**A**) Template quinazoline and (**B**) benzoxazole core of TLR7 inhibitors. (**C**) The alignment results of 54 TLR7 inhibitors was based on a field fit method.

**Figure 24 molecules-27-04026-f024:**
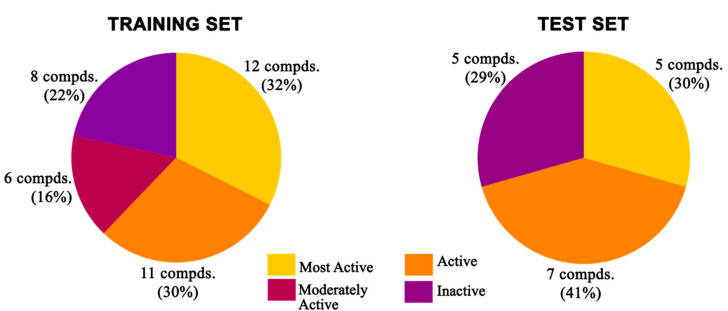
Distribution of different categories of compounds in training and test sets for 3D-QSAR study.

**Table 1 molecules-27-04026-t001:** Statistical outcomes of the top 10 pharmacophore hypotheses generated by the HypoGen algorithm.

Hypo. No.	Total Cost	Cost Difference	RMSD	Correlation	Max. Fit	Features
1	141.90	89.77	0.87	0.94	5.40	HBA, HYA, PI, RA
2	146.16	85.50	0.99	0.92	5.29	HBA, HBA, HYA, PI
3	146.47	85.19	1.02	0.92	5.77	HBA, HYA, PI, RA
4	148.27	83.40	1.03	0.92	6.10	HBA, HBA, HYA, PI, PI
5	148.27	83.40	1.03	0.92	5.01	HBA, HBA, HYA, PI
6	149.04	82.63	0.99	0.92	4.00	HYA, PI, PI, RA
7	149.04	82.62	1.05	0.91	4.93	HBA, HYA, PI, RA
8	149.11	82.55	1.07	0.91	5.34	HBA, HYA, PI, RA
9	149.13	82.53	0.97	0.93	3.78	HYA, PI, PI, RA
10	149.48	82.19	1.03	0.92	5.55	HBA, HBA, HYA, PI, PI

The table lists all the generated pharmacophore models and indicates their corresponding features that positively formed their properties and structural bases. The best model, Hypo1, was chosen based on cost, correlation, RMSD, and fit value for further studies.

**Table 2 molecules-27-04026-t002:** Experimental and estimated activity values of the training set compounds based on best pharmacophore hypothesis, Hypo1.

Comp No.	IC_50_ (μM)		Errors ^a^	Fit Value ^b^	Activity Scale ^c^	
	Experimental	Estimated			Experimental	Estimated
32	0.43	0.63	+1.47	5.06	++++	++++
33	0.5	0.75	+1.5	4.98	++++	++++
14	0.7	1.7	+2.43	4.64	++++	++++
38	0.8	0.73	−1.1	5.00	++++	++++
36	0.98	0.91	−1.08	4.92	++++	++++
35	0.99	0.73	−1.36	4.99	++++	++++
37	1.14	0.84	−1.36	4.95	++++	++++
19	1.2	3.1	+2.58	4.37	++++	+++
39	1.4	3.1	+2.21	4.88	++++	+++
13	1.4	1	−1.4	4.37	++++	++++
34	1.55	0.99	−1.57	4.88	++++	++++
15	4.4	8.3	+1.89	3.95	+++	+++
53	4.57	5.8	+1.27	4.11	+++	+++
48	4.71	5.4	+1.15	4.12	+++	+++
24	4.9	5.6	+1.14	4.13	+++	+++
50	4.99	5.7	+1.14	4.10	+++	+++
17	5.4	7.3	+1.35	4.01	+++	+++
22	5.7	8.1	+1.42	3.96	+++	+++
12	5.8	2.7	−2.15	4.44	+++	+++
52	8.09	7.8	−1.04	3.98	+++	+++
HCQ	8.2	17	+2.07	3.62	+++	++
49	8.3	11	+1.33	3.83	+++	++
21	8.7	4.1	−2.12	4.25	+++	+++
18	9.6	6.9	−1.39	4.03	+++	+++
16	11	8.6	−1.28	3.94	++	+++
26	16	8.1	−1.98	3.97	++	+++
27	17	7.6	−2.24	3.99	++	+++
25	17	7.5	−2.27	3.97	++	+++
7	20.7	37	+1.79	3.30	+	+
5	22	9.1	−2.42	3.91	+	+++
4	31	31	+1	3.36	+	+
45	37	160	+4.32	2.67	+	+
1	53	37	−1.43	3.30	+	+
41	185	190	+1.03	2.60	+	+
42	253	530	+2.09	2.15	+	+
43	272	160	−1.7	2.66	+	+
44	684	180	−3.8	2.62	+	+

^a^ Error factor was calculated as the ratio of the measured activity to the estimated activity in a way that the higher activity value was in the numerator and the smaller in the denominator; a positive value indicates that the estimated IC_50_ was higher than the experimental IC_50_; a negative value indicates that the estimated IC_50_ was lower than the experimental IC_50_ value. ^b^ Fit value indicates how well the features in the pharmacophore map fit with the chemical features present in the compound. ^c^ Activity scale: ++++, IC_50_ ≤ 2 μM (most active); +++, IC_50_: 2 to 10 μM (active); ++, IC_50_: 10 to 20 μM (moderately active); and +, IC_50_ > 20 μM (inactive).

**Table 3 molecules-27-04026-t003:** Experimental and estimated activity values of the test set compounds based on the Hypo1 pharmacophore.

Comp No.	IC_50_ (μM)		Errors ^a^	Activity Scale ^b^	
	Experimental	Estimated		Experimental	Estimated
31	0.46	0.92	+2.01	++++	++++
28	1.03	10.47	+10.17	++++	++
23	1.2	4.27	+3.55	++++	+++
9	1.3	8.69	+6.68	++++	+++
2	1.4	8.69	+6.21	++++	+++
29	1.83	5.89	+3.22	++++	+++
30	2.16	7.59	+3.63	+++	+++
20	4.2	6.17	+1.47	+++	+++
6	4.6	8.71	+1.89	+++	+++
11	5.6	6.61	+1.18	+++	+++
51	7.55	5.62	−1.34	+++	+++
10	11	39.81	+3.62	++	+
8	17	38.90	+2.29	++	+
40	23	157.04	+6.83	+	+
3	52	36.31	−1.43	+	+
47	110	190.55	+1.73	+	+

^a^ Error factor was calculated as the ratio of the measured activity to the estimated activity in a way that the higher activity value was in the numerator and the smaller in the denominator; a positive value indicates that the estimated IC_50_ was higher than the experimental IC_50_; a negative value indicates that the estimated IC_50_ was lower than the experimental IC_50_ value. ^b^ Activity scale: ++++, IC_50_ ≤ 2 μM (most active); +++, IC_50_: 2 to 10 μM (active); ++, IC_50_: 10 to 20 μM (moderately active); and +, IC_50_ > 20 μM (inactive).

**Table 4 molecules-27-04026-t004:** Structures of newly designed molecules and their predicted activities based on 2D and 3D-QSAR models.

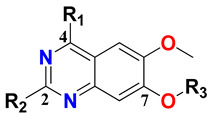
Comp.	R1	R2	R3	2D-QSAR	3D-QSAR	Pharmacophore
pIC_50_(µM)	IC_50_(µM)	pIC_50_(µM)	IC_50_(µM)	IC_50_(µM)
**T55**	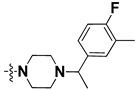	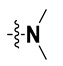	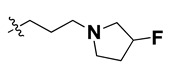	−0.40	2.50	−0.09	1.24	1.55
**T56**	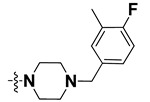	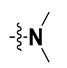	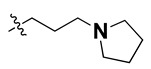	−0.19	1.55	−0.08	1.20	1.97
**T57**	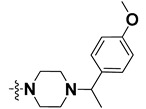	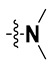	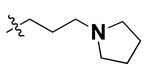	−0.24	1.73	0.01	0.98	1.51
**T58**	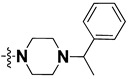	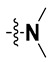	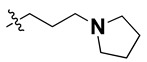	−0.48	3.04	−0.03	1.07	1.73
**T59**	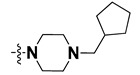	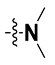	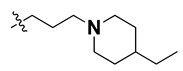	0.20	0.63	−0.20	1.58	1.78
**T60**	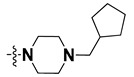	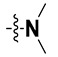	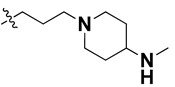	0.20	0.63	−0.10	1.25	1.50
**T61**	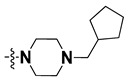	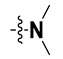	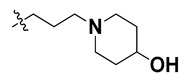	0.16	0.69	0.03	0.94	1.65
**T62**	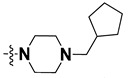	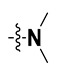	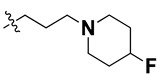	0.16	0.69	−0.08	1.21	1.42
**T63**	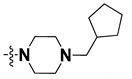	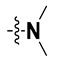	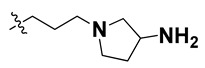	−0.42	2.62	0.00	1.01	1.62
**T64**	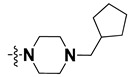	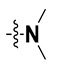	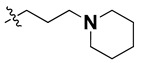	0.09	0.81	−0.19	1.53	1.39
**T65**	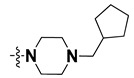	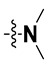	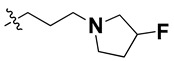	−0.42	2.62	−0.06	1.15	2.10
**T66**	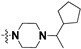	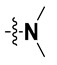	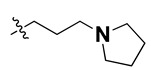	−0.50	3.15	0.07	0.85	1.82

**Table 5 molecules-27-04026-t005:** Binding analysis of newly designed small-molecule TLR7 antagonists.

Comp.	Antagonist Structure	Interacting Residue	Hydrogen Bond Formed	H-Bond Distance (Å)	Docking Score (kcal/mol)
**T55**	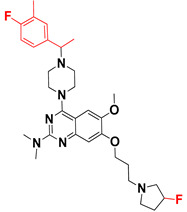	Lys432*, Gln324, Gly577, Tyr356*, Ser523	B:Lys432:HZ2–F38:T55 H44: T55O:Gln354:B T55:H60O:Gly577:A	2.02	123.23
**T56**	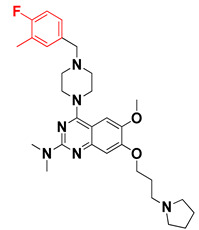	Gln354*, Val355*, Gly577, Lys432*	B:Lys432:HZ2F38: T56 T56:H42O:Gln354:B T56:H42O:Val355:B T56:H58O:Gly577:A	1.77	124.08
**T57**	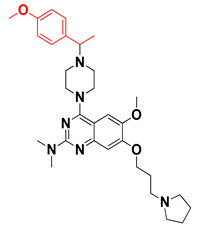	Gln354*, Val355*, Gly577, Phe500	T57:H43–O:Gln354:B T57:H43–O:Val355:B T57:H59–O:Gly577:A	1.85	114.97
**T58**	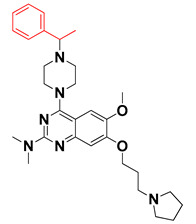	Gln354*, Val355*, Gly577	T58:H41–O:Gln354:B T58:H41–O:Val355:B T58:H57–O:Gly577:A	1.81	110.54
**T59**	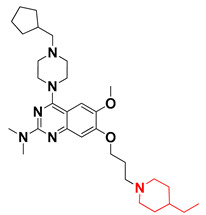	Thr525, Gln354*, Thr579, Tyr356*	T59:H42–OE1:Gln354:B T59:H42–O:Gln354:B T59:H58–OG1:Thr579:A T59:H77–OG1:Thr525:A	1.81	115.88
**T60**	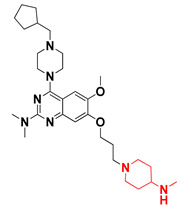	Gln354*, Thr525, Gly577	T60:H42–O:Gln354:B T60:H63–OG1:Thr525:A T60:H89–O:Gly577:A	2.53	133.30
**T61**	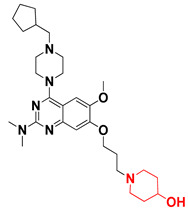	Gln354*, Thr525, Gly577, Tyr356*	T61:H41–O:Gln354:B T61:H62–OG1:Thr525:A T61:H87–O:Gly577:A	1.74	111.78
**T62**	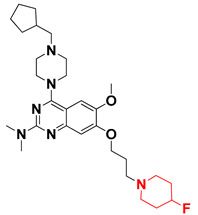	Gln354*, Gly577, Thr525, Asn255*, Tyr356*	T62:H41–B:Gln354:O T62:H57–A:Gly577:O T62:H62–A:Thr525:OG1	2.41	88.21
**T63**	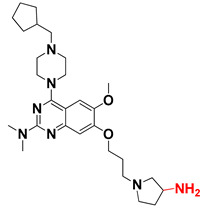	Gln354*, Thr525, Gly577, Tyr356*	T63:H40–B:Gln354:O T63:H61–A:Thr525:OG1 T63:H85–A:Gly577:O	2.16	108.50
**T64**	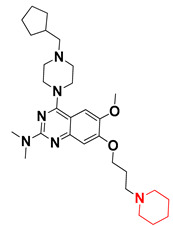	Gln354*, Thr525, Gly577, Tyr356*	T64:H40–O:Gln354:B T64:H56–O:Gly577:A T64:H61–OG1:Thr525:A	2.98	110.24
**T65**	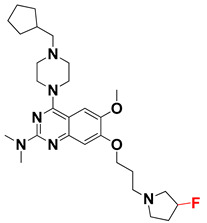	Gln354*, Thr525, Tyr356*, Thr406*	T65:H40–O:Gln354:B T65:H61–OG1:Thr525:A	1.82	112.45
**T66**	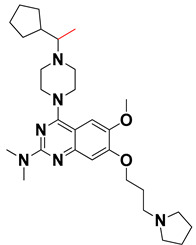	Gln354*, Gly577, Thr525, Tyr356*	T66:H40–O:Gln354:B T66:H50–O:GLY577:A T66:H63–A:Thr525:OG1	2.47	116.22

* represents the residues of chain B of the homodimeric protein.

**Table 6 molecules-27-04026-t006:** In silico ADMET prediction of newly designed compounds and the most potent template molecule.

Comp.	Absorption Level	AlogP98	PSA	BBB	BBB Level	Solubility	Solubility Level	Hepato- Toxicity	CYP2D6	CYP2D6 Probability
**14**	0	5.091	53.79	0.57	1.00	−5.71	2	False	−1.98	False
**T55**	1	5.861	53.79	0.81	0.00	−6.00	1	False	−0.78	False
**T56**	0	5.634	53.79	0.74	0.00	−5.94	2	False	−0.81	False
**T57**	0	5.303	62.72	0.49	1.00	−5.49	2	False	0.52	True
**T58**	0	5.32	53.79	0.64	1.00	−5.79	2	False	1.32	True
**T59**	1	6.256	53.79	-	4.00	−6.31	1	False	−3.40	False
**T60**	0	4.205	66.60	0.09	1.00	−4.89	2	False	−1.58	False
**T61**	0	4.063	74.61	−0.08	2.00	−4.45	2	False	−2.75	False
**T62**	0	5.003	53.79	0.54	1.00	−5.44	2	False	−2.28	False
**T63**	0	3.711	80.33	−0.28	2.00	−5.18	2	False	−2.64	False
**T64**	0	5.548	53.79	0.71	0.00	−6.01	1	False	−1.26	False
**T65**	0	4.941	53.79	0.52	1.00	−5.51	2	False	−2.91	False
**T66**	0	5.469	53.79	0.69	1.00	−6.03	1	False	−0.70	False

**Table 7 molecules-27-04026-t007:** Drug-likeness properties of newly designed inhibitors and the most potent template molecule.

Molecule	Molecular Weight (g/mol)	LogP	H-Bond Donors	H-Bond Acceptors	Number of Rotatable Bond	Polar Surface Area
**14**	496.69	5.09	0	8	10	57.2
**T55**	568.70	5.86	0	8	10	57.2
**T56**	536.68	5.63	0	8	10	57.2
**T57**	548.72	5.30	0	9	11	66.43
**T58**	518.69	5.32	0	8	10	57.2
**T59**	538.77	6.26	0	8	11	57.2
**T60**	539.76	4.21	1	9	11	69.23
**T61**	526.71	4.06	1	9	10	77.43
**T62**	528.71	5.00	0	8	10	57.2
**T63**	511.70	3.71	1	9	10	83.22
**T64**	510.72	5.55	0	8	10	57.2
**T65**	514.68	4.94	0	8	10	57.2
**T66**	510.72	5.47	0	8	10	57.2

**Table 8 molecules-27-04026-t008:** Toxicity results of the newly designed inhibitors and the most potent template molecule.

Comp.	FDA Carcinogenicity		FDA Carcinogenicity		AMES Mutagenicity	Rat oral LD_50_ (mg/kg)	Skin Irritation	Probability of Biodegradability
	Male Mouse	Female Mouse	Male Rat	Female Rat				
**14**	Non- carcinogen	Non- carcinogen	Non- carcinogen	Non- carcinogen	Non- mutagen	81.63	None	Non- degradable
**T55**	Non- carcinogen	Non- carcinogen	Non- carcinogen	Non- carcinogen	Non- mutagen	9.70	None	Non- degradable
**T56**	Non- carcinogen	Non- carcinogen	Non- carcinogen	Non- carcinogen	Non- mutagen	53.89	None	Non- degradable
**T57**	Non- carcinogen	Non- carcinogen	Non- carcinogen	Non- carcinogen	Non- mutagen	171.96	None	Non- degradable
**T58**	Non- carcinogen	Non- carcinogen	Non- carcinogen	Non- carcinogen	Non- mutagen	73.78	None	Non- degradable
**T59**	Non- carcinogen	Non- carcinogen	Non- carcinogen	Non- carcinogen	Non- mutagen	65.11	Mild	Non- degradable
**T60**	Non- carcinogen	Non- carcinogen	Non- carcinogen	Non- carcinogen	Non- mutagen	36.50	Mild	Non- degradable
**T61**	Non- carcinogen	Non- carcinogen	Non- carcinogen	Non- carcinogen	Non- mutagen	79.04	Mild	Non- degradable
**T62**	Non- carcinogen	Non- carcinogen	Non- carcinogen	Non- carcinogen	Non- mutagen	17.71	Mild	Non- degradable
**T63**	Non- carcinogen	Non- carcinogen	Non- carcinogen	Non- carcinogen	Non- mutagen	44.17	Mild	Non- degradable
**T64**	Non- carcinogen	Non- carcinogen	Non- carcinogen	Non- carcinogen	Non- mutagen	101.52	None	Non- degradable
**T65**	Non- carcinogen	Non- carcinogen	Non- carcinogen	Non- carcinogen	Non- mutagen	13.12	Mild	Non- degradable
**T66**	Non- carcinogen	Non- carcinogen	Non- carcinogen	Non- carcinogen	Non- mutagen	42.90	None	Non- degradable

## Data Availability

Data are available in the manuscript.

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
