# Peer review of "Integration of Ligand-Based and Structure-Based Methods for the Design of Small-Molecule TLR7 Antagonists"

_molecules, 2022, doi:10.3390/molecules27134026_

Round 1
Reviewer 1 Report
Dear Authors,
You will find a long list of remarks/questions below, however, your manuscript is interesting. I do prefer more technical information and then I think the manuscript stands a chance.
Please respond to:
1) Line 122: What was the 2D structural feature that you refer to?
2) Line 123: What was the descriptors and why were they selected as descriptors?
3) LIne 123: although I know that MLR is the abbreviation for multiple linear regression, you should define the abbreviation.
4) Eq (1) - terms of the equation should be defined.
5) Lines 128-129: define the terms. Also correct R2 to R2
6) Line 131 Define the term Q2LOO
7) Line 135-136: What is the limit for overfitting - ref [37] would have mentioned a specific value and also a reason why this value is considered to be the limit.
8) Figure 2 - how was the test set selected and why were 16 samples used for the test set?
9) Figure 3: The molecular descriptors need to be defined. The reader will not understand the abbreviations.
10) Line 154: How did you decide to do 2000 iterations and not more or less? What was you convergence criteria?
11) LIne 160-161: You need to define the abbreviations.
12) Line 168: Why is the threshold-value 0.6? and why are the two limits in line 169 set as acceptable limits? Was a reason for this provided in ref [41]?
13) Table 1: This is the last time I mention abbreviations. Somewhere, early in the text or in a list of abbreviations, all abbreviations should be defined for easy reference by the reader.
14) The sentence starting in line 183: "Thus, lesser..." needs to be rewritten.
15) Regarding line 183-184: why would atom properties not affect topology? What if a heavy atom is present instead of for example hydrogen.? Won't this affect the topology?
16) Line 190: What is meant by "intrinsic state"? Ref [46] must have defined it.
17) Line 202-203: Why is a logP of 0-0.25 a criterion? Why not other logP values? How would one administer the potential drug? I.e. orally, parenterally, topical etc? How would the logP affect the actual absorption through these routes of administration? It is true that activity may require a certain logP, but drug delivery may require something totally different.
18) Line 208 and rest of text: It is "Van der Waals" not "Van der Waal".
19) Wouldn't it be a stricter/better criteria to use a 2σ threshold for the Williams plot, Figure 5? You'd be able to narrow the standardized residual band quite a bit.
20) Line 243: Please motivate the choice of 37 compounds. There has to be some selection limit or percentage or reason to select n compounds.
21) Line 244: The abbreviations should be defined at the first instance where they are used.
22) Move the text in lines 260-266 before Table 2. The Figure caption is followed directly by Table heading. Place text in between the caption and heading. This will also make the proper reference to Table 2 possible.
23) Table 2: The "Features" column. Were those features only present in the fitted model or did they all contribute positively to activity? Perhaps you can make a footnote beneath the table with some information. For example, in all pharmacophore models, HBA had a negative impact on activity or whatever is applicable.
24) Line 272: "predictability power" should be "predictive power".
25) Line 275: Please replace the word "nicely" with something more formal. Yes, the meaning is clear, however, "nicely" is not a good word choice.
26) See (22) regarding the caption of Figure 7 being directly followed by Table 3.
27) Line 299: What is meant by "cost"? Is it computational cost, or does it mean the extent of changes that needed to be made to the structure to match the desired activity of the pharmacophore?
28) Line 314: Again motivate why 24 test compounds were used. Why 24?
29) Line 372 - the 70%/30% ratio must have been mentioned much earlier. There is also a lot of literature that explains why a 70/30 ratio is used.
30) If possible, cite a reference for the Insubria plot and make note that the Insubria plot is a variation of the Williams plot and is a plot to predict results for chemicals that have no experimental data available.

Author Response
A separate file is attached with the point-by-point response.

Reviewer 2 Report
The authors presented development of QSAR, pharmacophore and molecular docking models for the design of new TLR7 inhibitors and prediction of their activity. In addition, in silico estimation of pharmacokinetic properites, drug likeness and toxicity estimation of designed derivatives was performed. The manuscript is well writen and could be accepted for publicaiton after following clarifications and corrections were made:
- Minor grammar and spelling errors were found. You should re-check your manuscript again and correct these mistakes
- Some constructions must be more formal (e.g., line 67: "Interestingly, different groups, including us,"; line 82: "Various groups, including ours")
- The title should be changed to cover all in silico techniques used in this study (not only QSAR was applied). E.g.,"Integration of ligand-based and structure-based methods for the development of ...")
- Although supplementary material is large, some fugures and tables from the main text of the manuscript should be moved to the supplementary material (e.g., Figure 1, Table 1 and Table 5)
- The authors should mention which method was used for geometry minimization of tested compounds prior to the QSAR studies (line 686)
Author Response

(The authors gave the same response as above.)

Reviewer 3 Report
The submitted manuscript entitled “QSAR assisted exploration of ligand-based insights for the design of small-molecule TLR7 antagonists” by Sourav Pal et al. reports a comprehensive ligand-based (2D/3D-QSAR) and structure-related (molecular docking/dynamics) study of the antagonistic activities as well as the specific binding pattern against the endosomal TLR7 target protein.
From my perspective, the reviewed paper appears to be gripping to the scientific community, because the conjugation of in silico methodologies contribute considerably to the computational chemistry. On the other hand, some major concerns prevent the manuscript from suggesting for publication in the current form. In my humble opinion, some valid points should be definitely addressed before publishing the manuscript in Molecules.
Minor issues:
1. The entire text of the manuscript should be definitely revised and corrected, because there are many errors (punctuation mistakes), for instance:
Whole text. Authors overuse the present perfect tenses. The simple past tense is better in the scientific text.
Page 1. Line 41. Word repetitions should be avoided.
Whole text. R2/R2test etc. should be rounded to two decimal places – the rest is meaningless.
Page 2. I would avoid the words “including us/ours, from our groups, this present work” etc. It is not necessary.
Pages 8-9. The description of 2D descriptors should be moved to the Methods.
Page 10. Log P or logP. Unify it, please.
Page 10. Line 205. Space is missing ‘[42]enlists’.
Page 11. Hypo1 or Hypo 1. Unify it, please.
Page 12. Line 270. It should read ‘Table 3’.
Page 21. Line 475. It should read ‘(compound)’.
Page 21. Line. 479. It should read ‘model’.
Page 22. Line 495. What does ‘*’ in the name of the amino acid mean?
Page 22. Lines 510, 514. It should read ‘blue’.
Page 25. Lines 556, 559, 561, 634. It should read ‘and’ – without bold.
Page 28. Figure 18. Too many repetition of word ‘respectively’.
Page 30. Line 620. Missing dot at the end of the sentence.
Page 34. Line 743. It should read ’70:30%’.
Page 37. The whole subsection 3.4.3 should be deleted – it is obvious and meaningless.
Page 38. Line 871. It should read ‘(Table S4)’.
Pages 40-46. References should be checked and corrected, for instance, Ref. 32.
Major issues:
- I do not see a difference between 3D-QSAR and pharmacophore modeling, because CoMFA (as a classical 3D-QSAR strategy produces the pharmacophore pattern). Please, explain it.
- In Table 3. Errors are calculated as ‘ the ratio of the measured activity to the estimated activity’, but it works only for negative values. Why?
- Page 16. The Authors described the splitting procedure of compounds into a training and a test set based on the variations in the activities (Table 5), but different distributions of molecules are shown in Figs 22-25 in 2D/3D methods. Why?
- Page 25. Table 7. Have Authors synthesized a new set of molecules as it is suggested in the Table caption.?
- Page 37. What kind of scoring function was used to specify the best pose? What criterion was used?
Author Response

(The authors gave the same response as above.)

Round 2
Reviewer 1 Report
Dear Authors,
I am satisfied with the responses. I am especially happy that you added the training:test ratio of 70:30. This is also the ratio that machine learning techniques such as artificial neural networks use. So that is very good.
Regarding the MD study: It is a normal phenomenon that the candidate drug would move out of the pocket. It will move back into the pocket after even longer MD simulations. This is in agreement with some molecular mechanism models that propose that a drug won't stay permanently at its site of action. It is working like an on/off switch and as you know elimination of the drug also takes place. Glad you did a longer simulation.
I recommend acceptance after a final language check if anything was missed.
Kind regards
Reviewer 2 Report
The authors corrected their manuscript according to my instructions and it can be accepted for publication
Reviewer 3 Report
The Authors have followed my remarks.